# Accurate predictions of SARS-CoV-2 infectivity from comprehensive analysis

Jongkeun Park[1†], WonJong Choi[1†], Do Young Seong[1†], Seungpil Jeong[1], Ju Young Lee[1], Hyo Jeong Park[1], Dae Sun Chung[1], Kijong Yi[2], Uijin Kim[3], Ga-Yeon Yoon[3], Hyeran Kim[4,5], Taehoon Kim[4,5], Sooyeon Ko[6], Eun Jeong Min[7], Hyun-Soo Cho[3], Nam-Hyuk Cho[4,5,8], Dongwan Hong[1,9,10,11,12]*

[1]Department of Medical Informatics, College of Medicine, The Catholic University of Korea, Seoul, Republic of Korea; [2]Graduate School of Medical Science and Engineering, Korea Advanced Institute and Technology, Daejeon, Republic of Korea; [3]Department of Systems Biology, College of Life Science and Biotechnology, Yonsei University, Seoul, Republic of Korea; [4]Department of Microbiology and Immunology, Seoul National University College of Medicine, Seoul, Republic of Korea; [5]Department of Biomedical Sciences, Seoul National University College of Medicine, Seoul, Republic of Korea; [6]School of Chemical and Biological Engineering, Seoul National University, Seoul, Republic of Korea; [7]Department of Medical Life Sciences, College of Medicine, The Catholic University of Korea, Seoul, Republic of Korea; [8]Seoul National University Bundang Hospital, Seongnam, Republic of Korea; [9]Precision Medicine Research Center, College of Medicine, The Catholic University of Korea, Seoul, Republic of Korea; [10]Cancer Evolution Research Center, College of Medicine, The Catholic University of Korea, Seoul, Republic of Korea; [11]CMC Institute for Basic Medical Science, the Catholic Medical Center of The Catholic University of Korea, Seoul, Republic of Korea; [12]INNOONE, Seoul, Republic of Korea

*For correspondence:
dwhong@catholic.ac.kr

†These authors contributed equally to this work

Competing interest: The authors declare that no competing interests exist.

## eLife Assessment

The study provides **valuable** insight into the biological significance of SARS-CoV-2 by using a series of computational analyses of viral proteins. While the evidence is **solid**, the reviewers noted a lack of clarity about the objectives of the analyses. While impactful for the field, the manuscript would benefit from improved presentation.

**Abstract** An unprecedented amount of SARS-CoV-2 data has been accumulated compared with previous infectious diseases, enabling insights into its evolutionary process and more thorough analyses. This study investigates SARS-CoV-2 features as it evolved to evaluate its infectivity. We examined viral sequences and identified the polarity of amino acids in the receptor binding motif (RBM) region. We detected an increased frequency of amino acid substitutions to lysine (K) and arginine (R) in variants of concern (VOCs). As the virus evolved to Omicron, commonly occurring mutations became fixed components of the new viral sequence. Furthermore, at specific positions of VOCs, only one type of amino acid substitution and a notable absence of mutations at D467 were detected. We found that the binding affinity of SARS-CoV-2 lineages to the ACE2 receptor was impacted by amino acid substitutions. Based on our discoveries, we developed APESS, an evaluation model evaluating infectivity from biochemical and mutational properties. In silico evaluation using real-world sequences and in vitro viral entry assays validated the accuracy of APESS and our discoveries. Using Machine Learning, we predicted mutations that had the potential to become more prominent. We created AIVE, a web-based system, accessible at https://ai-ve.org to provide

infectivity measurements of mutations entered by users. Ultimately, we established a clear link between specific viral properties and increased infectivity, enhancing our understanding of SARS-CoV-2 and enabling more accurate predictions of the virus.

## Introduction

The importance of big data science has led to the accumulation of huge amounts of data, recently, a generative AI model with a large language model (LLM) using various types of big data has further been emerging across various fields, including healthcare, omics research, and industry (*Thirunavu-karasu et al., 2023*; *Yang et al., 2022*). In response to the continuous emergence of new SARS-CoV-2 variants, our study introduces an innovative model and platform that integrates big data analysis, protein structure prediction using AlphaFold, and AI learning to predict highly transmissible novel SARS-CoV-2 variants. The validation of predictions made using AI is crucial, so we present the proofs through in-silico and in-vitro experiments for highly infectious variants.

The COVID-19 pandemic-related deaths have decreased, but the infectivity of the severe acute respiratory syndrome coronavirus 2 (SARS-CoV-2) persists. This may be because of neutralizing antibodies from previous infections or numerous mutations in the virus, from the Alpha variant to current Omicron sublineages (*Tsai et al., 2021*; *Cao et al., 2022*).

Genomic databases, such as Nextstrain and the Global Initiative on Sharing All Influenza Data (GISAID), Our World in Data (OWID), containing epidemiological data, have been instrumental in collecting valuable data during the COVID-19 pandemic (*Shu and McCauley, 2017*; *Mathieu et al., 2021*; *Hadfield et al., 2018*). Reports on the current global efforts of genomic surveillance strategies and sequencing show varied levels of data accumulation (*Chen et al., 2022*). Nonetheless, the increasing amount of available SARS-CoV-2 data has made various analytical approaches possible.

SARS-CoV-2 mutations can alter gene function. For example, the SARS-CoV-2 furin gene plays an important role in the cleavage of the spike protein, and mutations in the gene significantly affect SARS-CoV-2 fusion with the host cell membrane (*Peacock et al., 2021*; *Johnson et al., 2021*). Mutations in the genes have a direct impact on its protein structure, influencing the pathway used by the virus to infect host cell (*Bouhaddou et al., 2023*). Mutations in the spike protein lead to differences in the protein structure and binding affinity to ACE2, the receptor through which the virus penetrates the host cell (*Ali et al., 2021*; *Seyran et al., 2021*).

Deep learning methods predict mutations and protein structures, and their application in research has led to enhanced genomic analysis with large datasets (*Senior et al., 2020*; *Theodosiou and Read, 2023*). This approach has proved valuable in predicting disease-related mutations and identifying genes of interest (*Yang et al., 2021*). Recently, deep learning has been utilized to analyze mutations in SARS-CoV-2 (*Berman et al., 2023*; *Zhou et al., 2023*). Prediction models, such as AlphaFold2, have been used to assess protein 3D structures (*Jumper et al., 2021*). This has led to the detection of structural changes caused by mutations in Delta (B.1.617.2) and Omicron (B.1.1.529) (*Bhowmick et al., 2022*). Alphafold2 has also been used to analyze protein genotypes and phenotypes in the RBD region (*Kilim et al., 2023*). Furthermore, hydrophobic properties in the amino acid sequence affect protein folding (*Lins and Brasseur, 1995*). Coronavirus hydrophobicity has significant effects on amino acid properties and protein folding (*Shekhawat and Roy Chowdhury Chakravarty, 2022*).

For these prior approaches to virus analysis and prediction, expertise in the relevant fields is required for a full understanding. Also, structure-based predictions of mutations cannot fully account for all aspects of SARS-CoV-2 infection. In this study, we analyzed SARS-CoV-2 mutations using artificial intelligence (AI) methods and leveraged large datasets to elucidate properties of the virus and make various predictions. We discovered amino acid polarity changes and substitutions and then evaluated the infectivity from significant mutations in the RBM region. We specifically examined the effect of protein structures from hydrophobicity to hydrophilic and alkaline properties.

Our evaluation involved a comprehensive analysis of epidemiological, and genomic data, capitalizing on the availability of large SARS-CoV-2 datasets. We extracted properties from the VOCs and each sublineage at the nucleotide and amino acid levels. We developed an evaluation model called amino acid property eigen selection score (APESS) to analyze SARS-CoV-2 sequences. Various methods were utilized to validate our findings, and we used machine learning to make further predictions.

Finally, we present our findings and evaluation models through the Artificial Intelligence Analytics Toolkit for predicting virus mutations in protEin (AIVE), a web-based platform that integrates SARS-CoV-2 data, visualizes APESS score distribution, offers 3D protein predictions, and increases accessibility for researchers. Overall, our research provides a model for pandemic preparedness and the study of infectious diseases.

## Results

### Discovery of significant properties in the amino acid sequence

We examined the amino acid sequences of SARS-CoV-2 to make discoveries about biochemical properties. We identified consecutive hydrophilic amino acids within the SARS-CoV-2 spike protein in the following lineages: Wuhan-Hu-1, Alpha, Beta, and Omicron (BA.1, BA.2, BA.2.75, BA.4/BA.5, XBB, BQ.1). A series of hydrophilic amino acid regions were observed in the RBM region (amino acid sequence 437–508). In specific SARS-CoV-2 lineages, amino acid substitutions were observed in these regions (*Figure 1A*, *Figure 1—figure supplement 1*). For example, the Delta and Omicron lineages contained a substitution of T478K of alkaline amino acids.

Each position showed differences in the number of polarity changes; specifically, changes from hydrophilic (polar, P) to hydrophobic (non-polar, N) or positively charged amino acids. For most positions, we did not detect significant differences in polarity between the lineages investigated (*Supplementary file 1a–c*).

As the virus progressed from the Wuhan-Hu-1 (40 in NN* and 31 in PP*) to Omicron (average 45.5 in NN* and average 27 in PP*), the number of polarity changes increased in NN* while decreasing in PP*. For these positions, polarity changes occurred in the lineages as they evolved to the Omicron variant (*Figure 1B*). More polarity changes occurred in Delta, XBB, Omicron BQ.1, and Omicron BA.4/BA.5 than in the other VOCs.

We investigated the amino acid substitutions in the VOCs and compared them with the Wuhan-Hu-1 sequence as the reference. We discovered a twofold increase in amino acid substitutions to lysine (K) and arginine (R) in later SARS-CoV-2 lineages. Meanwhile, glutamine (Q), phenylalanine (F), and glutamic acid (E) levels decreased by half in the VOCs. For phenylalanine (F), hydrophobic residue mutations occurred in areas irrelevant to the regions of consecutive hydrophilic amino acids (*Figure 1C* and *Supplementary file 1d* and *Figure 1—figure supplement 2*).

Various coronaviruses, including MERS-CoV, SARS-CoV-1, and SARS-CoV-2, did not show significant differences in polarity across positions (*Figure 1—figure supplement 3A*).

We investigated the polarity (hydrophilic, hydrophobic, alkaline, and acidic) of amino acids in Wuhan-Hu-1, Alpha, Beta, Omicron (BA.1, BA.2, BA.2.75, BA.4/BA/5, XBB, BQ.1), and the number of amino acids in the RBM sequence. Most amino acids were either hydrophilic or hydrophobic across lineages, with a slight increase in alkaline amino acid levels in the Omicron variants than in the others (*Figure 1—figure supplement 3B*).

We investigated the nucleotide sequences of the SARS-CoV-2 spike protein gene. We analyzed the transitions and transversions of 7,335,614 samples from the GISAID viral sequence. The rate of change from G to U was higher than that of U to G, and the rate from C to U was higher than that of U to C. Both C to G and G to C rates were extremely low (*Figure 1—figure supplement 3C* and *Supplementary file 1e*, *Panchin and Panchin, 2020*; *Yi et al., 2021*).

### Identification of properties in amino acid substitutions

We studied each mutation in the RBM region of lineages and sublineages using verified mutation data: Alpha, Beta, Delta, Omicron (BA.2.75), Omicron (BA.4/BA.5), Omicron (XBB), and Omicron (BQ.1).

From the viral sequences submitted to GISAID, we extracted data (n=7,335,614) of the sublineages defined on the outbreak.info platform. The most noteworthy mutations were N440, L452, S477, T478, E484, F486, N501, and Y505 (*Figure 2A*). Only seven individuals possessed mutations at D467 out of the 7,335,614 investigated (*Figure 2—figure supplement 1A*).

Importantly, the experimental results corroborated the amino acid substitutions observed in the viral sequences. Almost no mutations were found at the D467 position of the RBM sequence. In vitro experiments using a luciferase assay and viral entry experiments demonstrated that infectivity

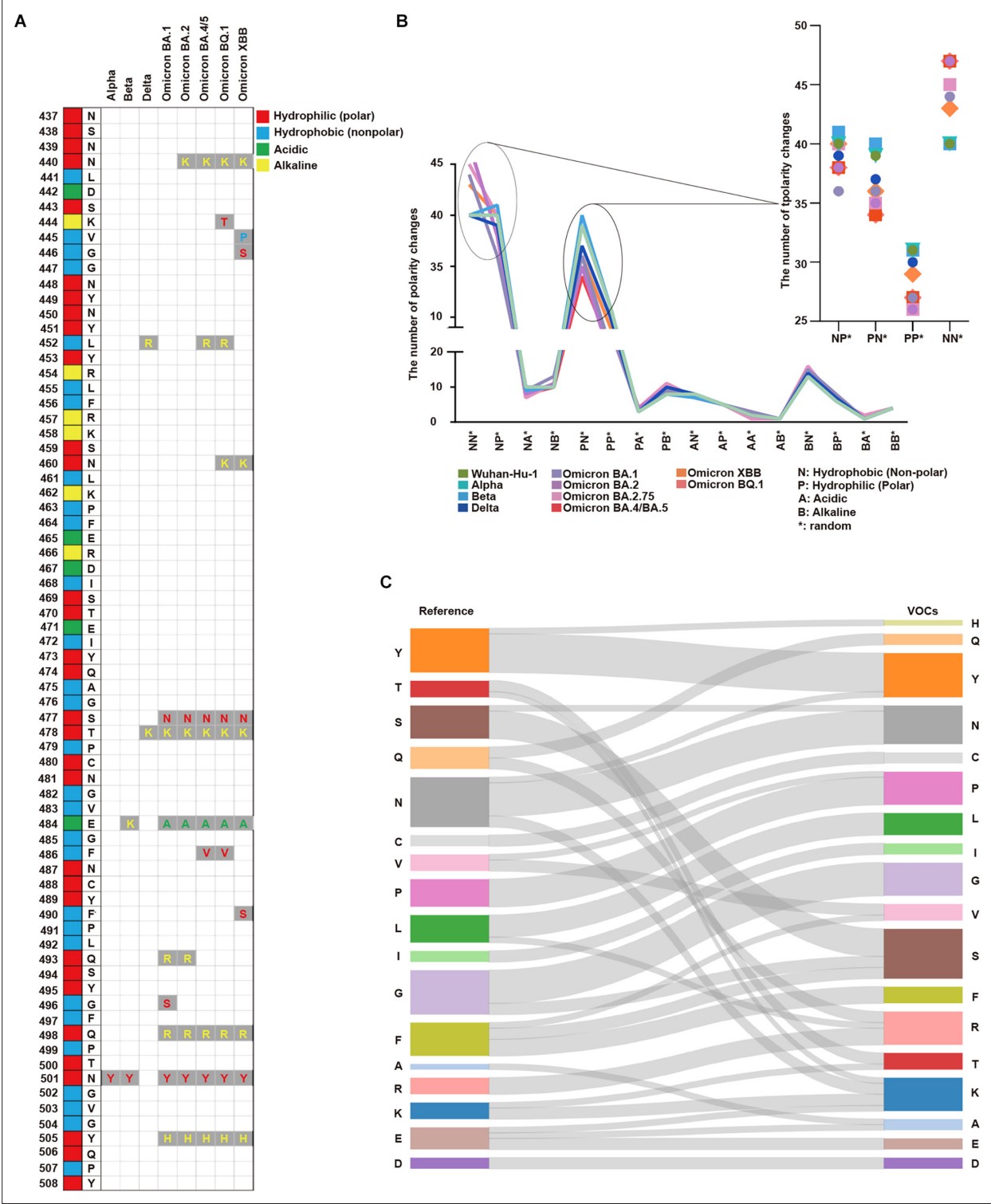

**Figure 1.** Analysis of protein properties discovered in the SARS-CoV-2 amino acid sequence. (**A**) The SARS-CoV-2 amino acid sequence between positions 437 and 508 in the receptor binding motif (RBM) is displayed with the corresponding amino acids in the original positions. Amino acid substitutions are shown for Alpha, Beta, Delta, Omicron BA.1, Omicron BA.2, Omicron BA.4/BA.5, Omicron BQ.1, and Omicron XBB. Hydrophilic (polar) amino acids are displayed in red, hydrophobic (non-polar) in blue, acidic in green, and alkaline (positively charged) in yellow. (**B**) The number of polarity changes [N: hydrophobic (nonpolar), P: hydrophilic (polar), A: acidic, and B: alkaline (basic)] in the receptor binding domain (RBD) region is displayed. Wuhan-Hu-1, Alpha, Beta, Delta, Omicron (BA.1), Omicron (BA.2), Omicron (BA.2.75), Omicron (BA.4/BA.5), Omicron (XBB), and Omicron (BQ.1) are presented on the graph with each lineage color-coded. The polarity change count for NN*, NP*, PN*, and PP* are shown in more detail. PN*

*Figure 1 continued on next page*

*Figure 1 continued*

(Wuhan-Hu-1: 39, Alpha: 39, Beta: 40, Delta: 37, Omicron BA.1: 36, Omicron BA.2: 35, Omicron BA.2.75: 35, Omicron BA.4/BA.5: 34, Omicron XBB: 36, Omicron BQ.1: 34) PP* (Wuhan-Hu-1: 31, Alpha: 31, Beta: 31, Delta: 30, Omicron BA.1: 27, Omicron BA.2: 26, Omicron BA.2.75: 26, Omicron BA.4/BA.5: 27, Omicron XBB: 29, Omicron BQ.1: 27). Overall, polarity decreased from PN* to PP* across all SARS-CoV-2 lineages. (**C**) The amino acid substitutions in the RBM region from the reference to VOCs are displayed. The seventeen amino acids in the reference list are tyrosine (**Y**), threonine (**T**), serine (**S**), glutamine (**Q**), asparagine (**N**), cysteine (**C**), valine (**V**), proline (**P**), leucine (**L**), isoleucine (**I**), glycine (**G**), phenylalanine (**F**), alanine (**A**), arginine (**R**), lysine (**K**), glutamic acid (**E**), and aspartic acid (**D**). For variants of concerns (VOCs), the amino acid substitutions are indicated by gray lines. There was a more than twofold increase in lysine (**K**) and arginine (**R**) in VOCs compared with the reference.

The online version of this article includes the following figure supplement(s) for figure 1:

**Figure supplement 1.** Research Overview.

**Figure supplement 2.** Amino acid substitutions in spike proteins from variants of concerns (VOCs).

**Figure supplement 3.** Polarity changes in the spike protein region of coronavirus and characteristics of RNA and amino acid levels of SARS-CoV-2.

**Figure supplement 4.** Polarity change analysis due to mutation in the amino acid sequence of MERS-CoV.

---

associated with D467 mutagenesis (D467P and D467I) was lower than the wild-type (spike protein D614G) (*Figure 2C*, *Figure 2—figure supplement 2*).

We investigated the mutation rates and amino acid substitutions in VOCs and variants under monitoring (VUMs). In the earlier Alpha, Beta, and Delta lineages, various amino acid changes occurred at multiple locations. In later lineages, we observed a reduction in the diversity of amino acid changes due to substitutions (*Figure 2B*). L452R and T478K were the most numerous non-synonymous mutations while more synonymous mutations developed.

## Differences in molecular cell levels that affect infectivity and severity in the evolution of SARS-CoV-2

We investigated the molecular cell level differences that affect the severity and infectivity of SARS-CoV-2 in its evolutionary process. From 1/23/2020 to 12/31/2022, we examined the number of infections and deaths using data from OWID and compared the periods where lineages occurred. Based on epidemiological data, we established three main periods: the major outbreak period of Delta, Omicron BA.5, and Omicron BQ. During the Delta period, there was an increase in the ratio of deaths to infections, for the Omicron BA.5 period, both infections and deaths increased, and finally in the Omicron BQ period, both decreased (*Figure 3A*). We attributed these trends to mutations in the major lineages and further investigation of sublineages through viral sequence information revealed stabilization per lineage (*Supplementary file 1f and g*).

We investigated the epidemiological relationship between symptoms and disease severity in patients by examining mutations in the RBM region for major SARS-CoV-2 lineages and their sublineages. We found that these mutations became fixed over time (*Figure 3B* and *Supplementary file 1h and* i). In the Delta variant, a significant increase in symptomatic infections occurred for mutations L452R (odds ratio 4.346, 95% CI 2.378–8.191) and T478K (odds ratio 3.116, 95% CI 1.595–6.292). For most patients with Omicron BA.5 and BQ.1, mutations were asymptomatic. However, among patients with Omicron BQ.1 mutations, only those with K447T were symptomatic. The Delta, Omicron BA.5, and Omicron BA.1 variants more frequently produced milder outcomes with more mutations. Consequently, we showed that as SARS-CoV-2 evolves to Omicron, the number of asymptomatic patients with mild outcomes increased, with infection symptoms becoming less severe.

Since we cannot determine the severity and presence of symptoms in SARS-CoV-2 from mutations alone, we analyzed gene expression within host cells of infected patients. We utilized GSE235262 from the NCBI GEO database to analyze the gene expression of controls (uninfected), Alpha (B.1.1.7), Delta (B.1.617.2), and Omicron (B.1.529, BA.2, BA.4, and BA.5) variants during waves of COVID-19. In particular, we focused on the expression differences between Alpha (B.1.1.7) and Omicron (BA.4/BA.5), and between Delta (B.1.617.2) and Omicron (BA.4/BA.5), conducting enrichment analyses on genes that showed significant differences (*Supplementary file 1j*). Compared to Omicron (BA.4/BA.5), most genes in Delta (B.1.617.2) were up regulated in molecular and virus infection pathways (p<0.05). Notably, in the mTOR pathway, there was a correlation with receptor genes leading to the PI3K-AKT pathway, lysosome, ribosome biogenesis, autophagy, and lipid biosynthesis, In the virus infection pathway, endocytosis and metabolic pathways were related (*Figure 3C*). Compared to

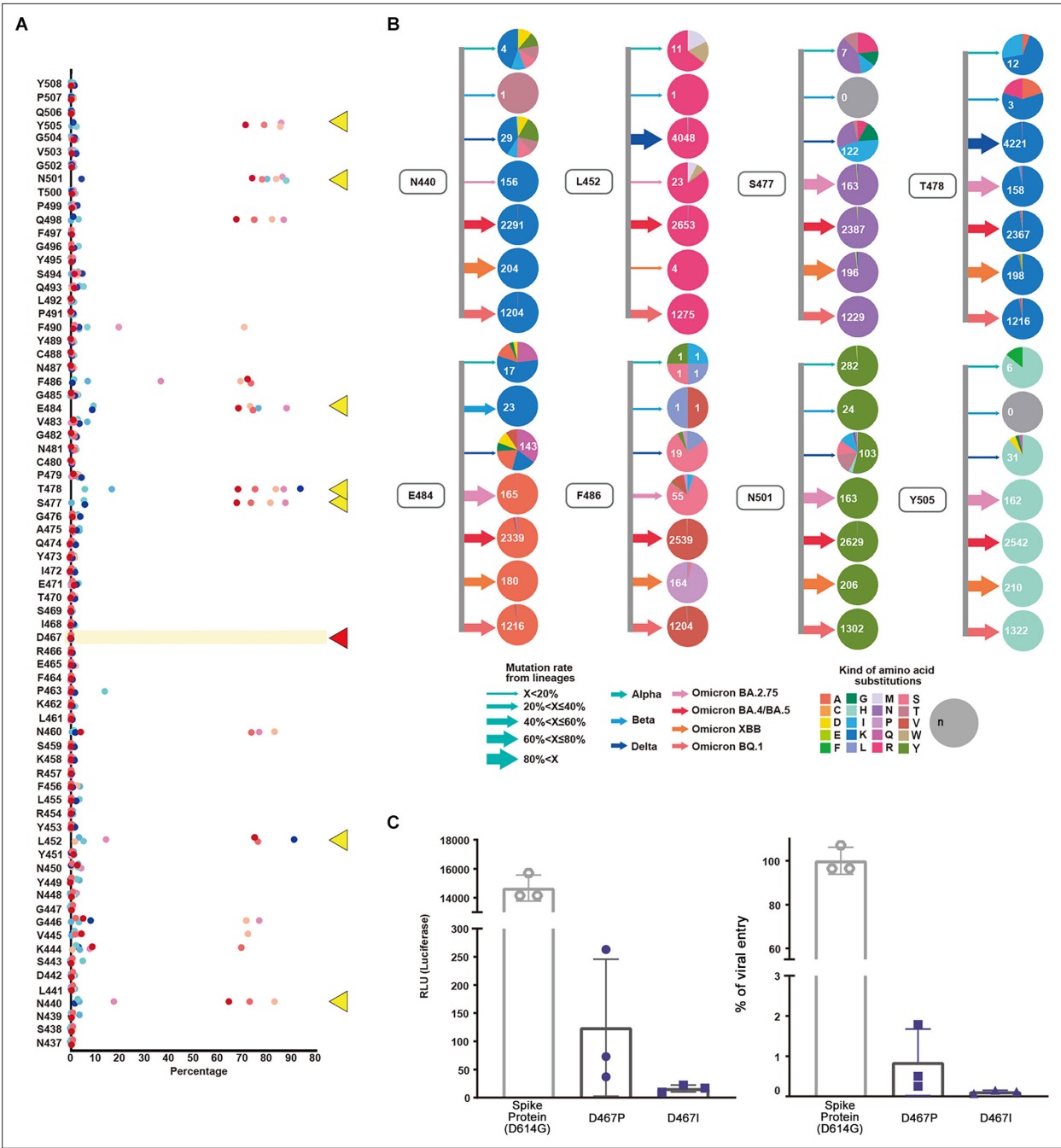

**Figure 2.** Evaluation and validation of amino acid substitutions in the SARS-CoV-2 receptor binding motif (RBM) region. (**A**) Mutations, their occurrence rates as percentages, and the original amino acid at the position are shown. Alpha, Beta, Delta, Omicron (BA.2.75), Omicron (BA.4/BA.5), Omicron (XBB), and Omicron (BQ.1) lineages are displayed with corresponding colors. N440, L452, S477, T478, E484, F486, N501, and Y505 are indicated by yellow triangles while D467 is indicated by a red triangle. (**B**) For positions N440, L452, S477, T478, E484, F486, N501, and Y505, lineages and amino acid substitutions are displayed. Arrows indicate the mutation rate where width corresponds with the percentage and the colors indicate lineages Alpha, Beta, Delta, Omicron (BA.2.75), Omicron (BA.4/BA.5), Omicron (XBB), and Omicron (BQ.1). The colors in the pie chart indicate amino acids. The mutation rate of Alpha is lower than 20%. At the 501st position in the RBM region, amino acid substitutions from asparagine (**N**) to tyrosine (**Y**) (n=282) occurred along with other substitutions. For the Beta variant, the 484[th] position showed a mutation rate over 60% with E484K. The Delta variant showed a mutation rate of 60% at the 452nd position. L452R and T478K amino acid substitutions along with various mutations were observed. In Omicron, the mutation rate for S477, T478, E484, N501, and Y505 was over 40%. The amino acid substitutions were S477K, T478K, E484A, N501Y, and Y505H. We calculated the mutation rates for the following positions: T478K (99.95%), Q498R (94.14%), N501Y (99.52%), and Y505H (97.66%). (**C**) The effect of the D467 amino acid substitution on viral infection was evaluated in vitro via luciferase and viral entry assays. Mutagenesis at D467 to hydrophobic amino

*Figure 2 continued on next page*

*Figure 2 continued*

acids proline (**P**) and isoleucine (**I**) was performed. There was a significant decrease in the RLU and viral entry percentages for both D467P and D467I (<0.0001).

The online version of this article includes the following figure supplement(s) for figure 2:

**Figure supplement 1.** Presence of D467 mutation from Global Initiative on Sharing All Influenza Data (GISAID) and evaluation of protein structure due to D467.

**Figure supplement 2.** Identification of protein expression using western blotting.

**Figure supplement 3.** D467 protein 3D structure prediction.

Omicron (BA.4/BA.5), Delta is more associated with various pathways within the host cell, leading to inflammation, replication, and severity.

We believe that Omicron's characteristics are not solely due to its molecular cell level features within the host cell but also stem from its evolution, affecting its binding affinity to the host's ACE2. Therefore, we utilized pDockQ and HADDOCK to predict the binding affinity between SARS-CoV-2 and the ACE2 receptor affected by mutations (*van Zundert et al., 2016*; *Bryant et al., 2022*). Based on the pDockQ results, with Wuhan-Hu-1 as the standard, the SARS-CoV-2 variants showed strong affinity in descending order of Delta (0.577), Omicron XBB (0.575), Beta (0.569), Omicron BQ.1 (0.564), and Omicron BA.4/BA.5 (0.560) (*Figure 3D*, *Figure 3—figure supplement 1*). Compared to the binding affinity of Wuhan-Hu-1 (0.506), the lineages with higher binding affinity were associated with infectivity (*Supplementary file 1k–1* m and *Figure 3—figure supplement 2*).

The results showed the following descending order of binding affinities for the ACE2 receptor: Delta, XBB, Beta, BQ.1, BA.4/BA.5, BA.1/BA.2, Alpha, and Wuhan-Hu-1. While Beta had a high-affinity score, it has had a low occurrence rate since its emergence, and it did not receive a high score from our evaluation model.

To check for interspecies infection in humans, bats, and pangolins, we measured the bond affinity between the virus and the host. We also measured the bond affinity of SARS-CoV-1. The bond strength of SARS-CoV-2 to the *Homo sapiens* ACE2 receptor is weaker than that of SARS-CoV-1 (*Cao et al., 2021*). Although the bond strength with the same host is higher, it weakens with a different host. This suggests that viruses that evolve within a human host are likely to be the most infectious to humans. As SARS-CoV-2 evolved to Omicron, the binding affinity increased for the virus.

Overall, the Delta variant was found to be more associated with the host molecular pathway and severity. Meanwhile, the Omicron variant showed a higher interaction between the ACE2 and RBM region. The Omicron variant is more infectious.

## Development of APESS, an evaluation scoring model and, the evaluation of lineages

We developed APESS, an evaluation model to analyze viral sequences based on the nucleotide, amino acid, and protein structure properties. APESS was calculated from four previous calculations: sub-clustering of protein structure (SCPS), polarity change score (PCS), mutation rate (MR), and biochemical properties eigen score (BPES) (*Figure 4A* and *Supplementary file 1n–q* and *Figure 4—figure supplement 1*). The detailed calculations and components of APESS are displayed in the Materials and Methods section.

We calculated the APESS scores of 7,335,614 viral sequences encompassing VOCs and their sublineages (*Figure 4B*). The Alpha and Beta variants received lower APESS scores, whereas the Delta and Omicron variants (BA.5, BQ.1, and XBB) received higher scores.

We generated an APESS distribution graph for the VOC sublineages. Significant variants, including Delta and Omicron, showed APESS values higher than 1.62, indicating high infectivity (*Figure 4A*, *Figure 4—figure supplement 2*). The APESS scores of the sublineages were similar to those of the VOCs. Some sublineages had lower APESS scores because of the wide variety of amino acid substitutions in the RBM region (*Figure 4B*). The sublineages with the highest APESS scores possessed amino acid substitutions to lysine (K) and arginine (R) at S477.

APESS evaluates the infectivity of the SARS-CoV-2 lineage. Lower APESS scores indicated that the lineage is less infectious, whereas higher APESS scores indicated that it may be more infectious.

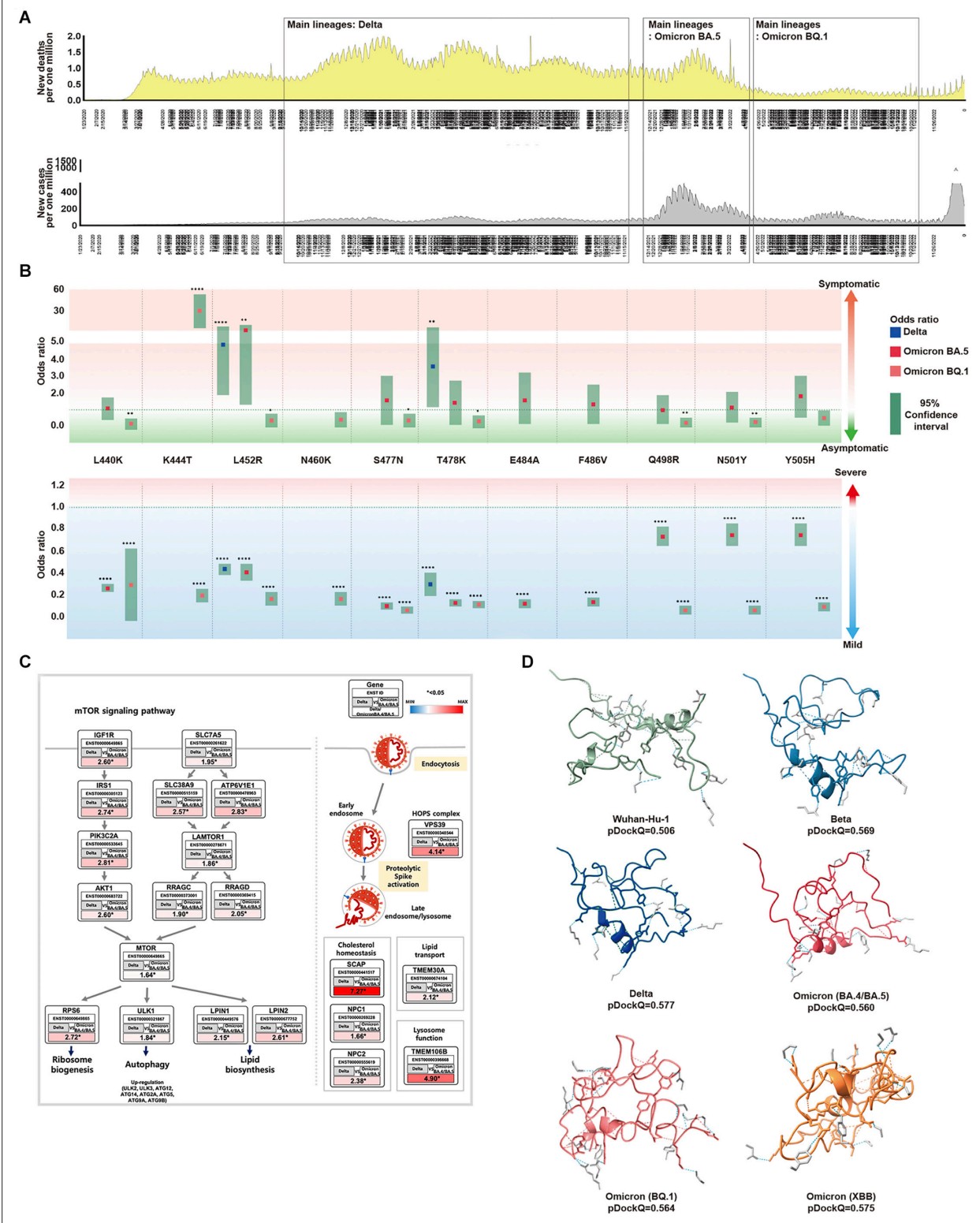

**Figure 3.** Association between SARS-CoV-2 mutations and epidemiological data. (**A**) For each SARS-CoV-2 lineage, the position and distribution of amino acid substitutions were analyzed alongside epidemiological data. The first period, from November 2020 to December 2021, was characterized by low infections and high deaths. Delta was the most prominent during this period, coinciding with worldwide vaccinations. Amino acid substitutions L452R and T478K were observed at the highest frequency while T478K and L452R were independently observed at the highest frequencies. The second period, from January 2022 to April 2022, saw an increase in the new cases and deaths. BA.5 was prominent during this period with various amino

*Figure 3 continued on next page*

*Figure 3 continued*

acid substitutions observed. The third period, from May 2022 to November 2022, showed a significant decrease in infections and deaths. Omicron, specifically BQ.1 was prominent during this period and worldwide vaccination rates decreased. (**B**) From the viral sequences of the patients, the association between the primary mutations of Delta, BA.5, and BQ.1, and epidemiological data (symptoms and severity) was analyzed. Odds ratios are displayed for L440K, K444T, L452R, N460K, S477N, T478K, E484A, F486V, Q498R, N501Y, and Y505H. L452R was an indicator of symptomaticity in Delta and BA.5. K477T was associated with symptomaticity in BQ.1. All mutations were associated with mildness. The 95% confidence intervals are shown for Delta, BA.5, and BQ.1. (**C**) We analyzed the expression of Delta compared to BA.4/BA.5 using the GSE235262 dataset. In Delta, the mTOR pathway was observed to regulate ribosome biogenesis, autophagy, and lipid biosynthesis while also playing a role in viral infection pathways. The expression values were calculated as Delta [log2(TPM +1)] / Omicron BA.4+BA.5 [log2(TPM +1)], with ENST Ensembl Transcript IDs, and * indicating a significance level of p<0.05. (**D**) The folding structures and pDockQ scores (0.506, 0.569, 0.577, 0.560, 0.564, and 0.575 for Wuhan-Hu-1, Beta, Delta, BA.4/BA.5, BQ.1, and XBB, respectively) were shown.

The online version of this article includes the following figure supplement(s) for figure 3:

**Figure supplement 1.** Prediction of spike protein 3D structures from Alpha Omicron BA.1 Omicron BA.2.

**Figure supplement 3.** pDockQ scores of Wuhan-Hu-1 mutations.

**Figure supplement 4.** pDockQ scores of Alpha mutations.

**Figure supplement 5.** pDockQ scores of Beta mutations.

**Figure supplement 6.** pDockQ scores of Delta mutations.

**Figure supplement 11.** pDockQ scores of BQ.1 mutations.

**Figure supplement 12.** pDockQ scores of XBB mutations.

Based on cumulative prevalence results from investigations on March 21, 2023 (outbreak.info), 2% cumulative prevalence of the S477K mutation was confirmed in the Omicron variants BA.5.1 and BA.5.2. Severe infections were associated with amino acid substitutions to lysine (K) and arginine (R) at the S477 position.

Utilizing a gaussian mixture model (GMM), we divided the APESS scores of 30,000 randomly selected sublineages into four components. Four groups were identified, G1 (Alpha and Beta: a centroid at 0.0289), G2 (BA.1, BA.2.75, and XBB: a centroid at 1.9474), G3 (Delta: a centroid at 1.6383). G4 (BA.2, BA.4/5 and BQ.1: a centroid at 2.0802). We also examined the distribution for all 7,335,614 sequences and found that the results were the same as the 30,000 sampled sublineages. We observed that most of Alpha and Beta were included in the component with a centroid of 0.0289, while the component with a centroid of 2.0627 included various Omicron variants.

## Creation of candidate lineages and protein structure predictions

We introduced amino acid substitutions at specific locations in the SARS-CoV-2 backbone for the wild-type and VOCs. The amino acid substitutions were lysine (K), arginine (R), asparagine (N), serine (S), tyrosine (Y), and glycine (G). We then evaluated the infectivity of these candidate lineages with our evaluation model APESS. APESS scores of candidate lineages with amino acid substitutions to positively charged residues K and R were higher than those of the SARS-CoV-2 variants. In particular, S477K and S477R amino acid substitutions were linked to increased infectivity. In contrast, amino acid substitutions of N, S, Y, and G resulted in APESS scores similar to or lower than of the SARS-CoV-2 variants (*Figure 5A*).

Of the mutated sequences, amino acid substitutions of K and R showed the highest APESS and pDockQ scores (*Figure 5B*).

We used deep learning and machine learning methods to determine the probability of amino acid substitutions at specific locations in the lineages before and after the Omicron variant emergence. Models such as random forest, light-gradient boosting machine (LightGBM), extreme Gradient Boosting (XGBoost), and ensemble methods were used, and consistent results were obtained (*Figure 5—figure supplement 1* and *Supplementary file 1r–1* v). Specifically, for N460K, we found

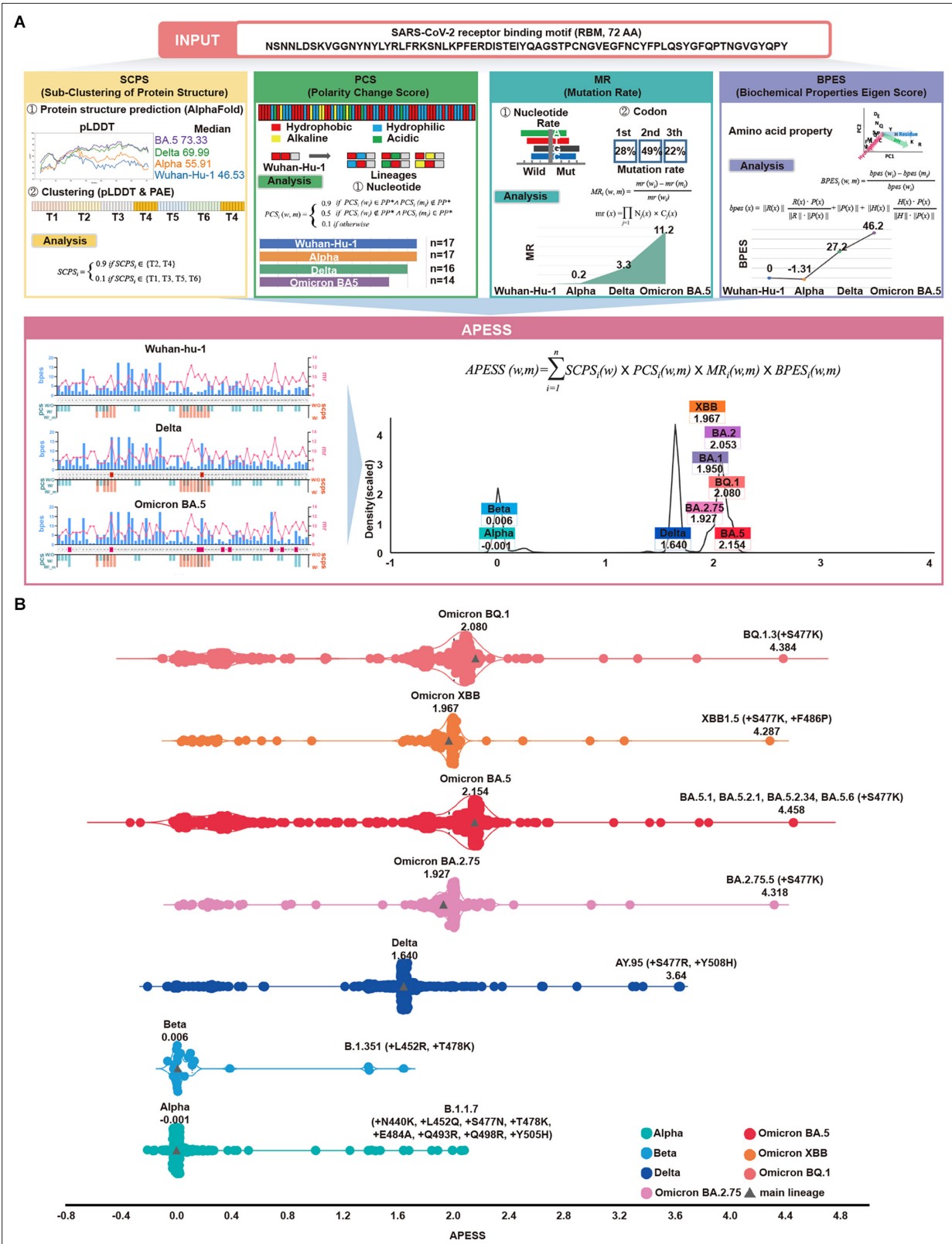

**Figure 4.** APESS: a comprehensive evaluation model of SARS-CoV-2 mutations. (**A**) Amino acid property eigen selection score (APESS), an evaluation model based on the properties discovered in the receptor binding motif (RBM) and the infectivity of SARS-CoV-2, was developed. A 72-amino acid-long RBM sequence of SARS-CoV-2 was used to comprehensively evaluate the sub-clustering of protein structure (SCPS), polarity change score (PCS), mutation rate (MR), and biochemical properties eigen score (BPES). Through comprehensive analysis of each position, the infectivity of the input

*Figure 4 continued on next page*

*Figure 4 continued*

sequence could be evaluated against preexisting lineages. (**B**) The APESS scores were calculated for SARS-CoV-2 lineages Alpha, Beta, Delta, and Omicron (BA.2.75, BA.5, XBB, BQ.1), and the data were obtained for sublineages from viral sequences. The original lineages are displayed with a gray triangle and their APESS scores, whereas the sublineages are color-coded differently. The S477K substitution resulted in the highest APESS score.

The online version of this article includes the following figure supplement(s) for figure 4:

**Figure supplement 1.** Amino acid property eigen selection score (APESS) evaluation of each section from variants of concerns (VOCs).

**Figure supplement 2.** Amino acid property eigen selection score (APESS) distribution from viral sequences and APESS scores of variants of concerns (VOCs) from Global Initiative on Sharing All Influenza Data (GISAID).

that the probability increased as the lineage progressed to Omicron. For Q493R, the probability was relatively constant for Omicron lineages (*Figure 5C* and *Supplementary file 1w*). We used Accuracy, Precision, Recall, and F1 score to evaluate performance. All models showed high-performance scores above 0.95 in Precision, Recall, and F1 score. For accuracy, XGBoost, scored above 0.89, exhibiting relatively high performance while LightGBM scored above 0.78.

To validate the binding kinetics of Q493R and N460K variant RBDs to human ACE2 at the macromolecular level, we performed surface plasmon resonance (SPR). In line with our computational prediction, both variants show approximately a threefold increase in binding affinity compared to the wild-type RBD (*Figure 5—figure supplements 2 and 3*).

We further verified the infectivity of the predicted mutations using luciferase and viral entry assays (*Figure 5D*). N437R mutation led to a twofold increase in luciferase activity and viral entry compared to the wild-type (spike protein D614G). In contrast, N460K and Q493R mutations led to a decrease in luciferase activity and viral entry compared to the wild-type (spike protein D614G) (*Figure 5D*, *Figure 5—figure supplement 3*). However, the viral entry of N460K was 10-fold higher than that of Q493R. N460K mutations occurred at a low rate but had a higher probability of infection compared to other mutations.

Therefore, our results imply that the maintenance of the proliferative rate of the virus was due to new mutations, high docking scores, and an increased presence of Omicron variants (*Figure 3—figure supplements 3–12*).

Based on our findings, we developed AIVE (https://ai-ve.org/), a web-based platform enabling rapid and precise prediction of protein structure and calculation of SARS-CoV-2 infectivity. AIVE facilitates the analysis of virus sequences entered by users and provides visualization and analysis reports, allowing it to be used in environments without GPU installation (*Figure 6*). For example, we used a customized sequence wherein N460K substitution was introduced into the Wuhan-Hu-1 sequence (Wuhan-Hu-1+N460 K) as an input. The output generated the following four results:

First, protein structure prediction results showed that Wuhan-Hu-1+N460 K had increased folding compared to Wuhan-Hu-1. Meanwhile, XBB, which contains N460K substitution, demonstrated a more stabilized protein structure compared to Wuhan-Hu-1+N460 K due to the alpha-helix in the 493–495 region and the beta-sheet in the 465–467 and 505–507 regions (Panel ① of *Figure 6*). Second, for polarity changes, there was a decrease in consecutive hydrophilic amino acids in Wuhan-Hu-1+N460 K due to the mutation (Panel ② of *Figure 6*). Third, calculations of the APESS sub-scores showed a difference in scores for XBB compared to Wuhan Hu-1 +N460 K owing to mutations (Panel ③ of *Figure 6*). Fourth, the APESS distribution was displayed as 'apess,' which reflects the score at each position in the amino acid sequence. The apess score at N460K for both Wuhan-Hu-1+N460 K and XBB (with N460K) was 0.112 (Panel ④–1 of *Figure 6*). Furthermore, the distribution of the total 'APESS' score, which comprehensively evaluates each position of the sequencing results, was used to determine the infectivity sections. Usage guidelines of AIVE can be found in supplementary information (*Figure 6—figure supplement 1*).

An apess score at N460K for XBB (with N460K) was 1.967. XBB (with N460K) mutation was located in the infectivity region with a statistically significant association (>95%, pink color), whereas APESS score increased for Wuhan Hu-1 +N460 K compared to Wuhan Hu-1 but was not included in the high infectivity section containing XBB (Panel ④–2 of *Figure 6*). Therefore, our research findings enable rapid protein structure prediction and APESS via meticulous and systematic structuring of data. Visualization of these analyses and the evaluation of infectivity are available on AIVE.

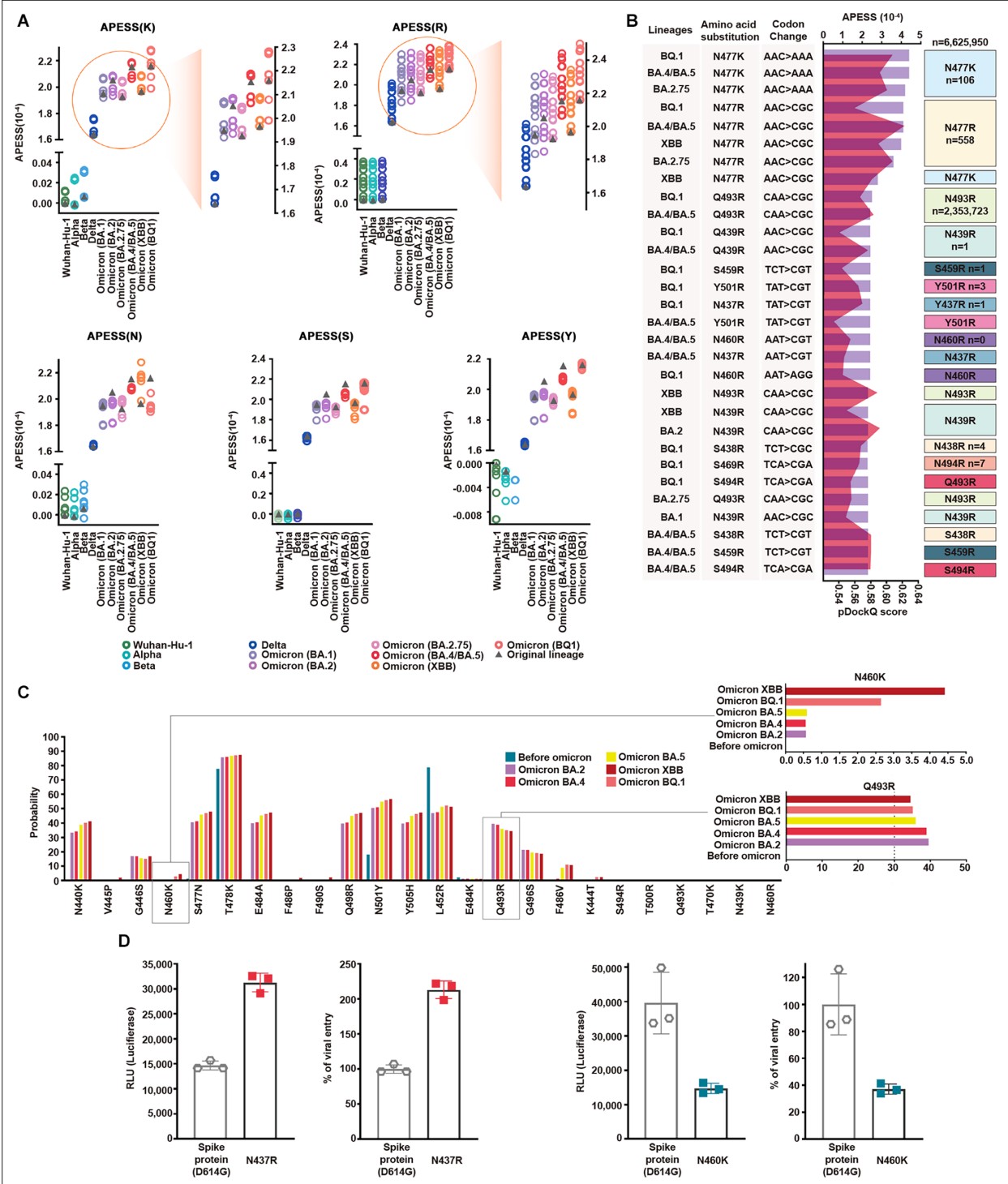

**Figure 5.** Multifaceted evaluation of SARS-CoV-2: evaluation model, machine learning, and in vitro assay. (**A**) Mutagenesis sequences containing consecutive hydrophilic amino acids were evaluated with amino acid property eigen selection score (APESS). They were based on Wuhan-Hu-1, Alpha, Beta, Delta, BA.1, BA.2, BA.2.75, BA.4/BA.5, XBB, and BQ.1, as indicated by colors and gray triangles. APESS values for the K, R, N, S, and Y mutated sequences of the lineages are displayed. Mutagenesis of lysine (**K**) and arginine (**R**) in Omicron sublineages resulted in increased APESS scores, whereas mutagenesis of asparagine (**N**), serine (**S**), and tyrosine (**Y**) resulted in decreased APESS scores. Specific regions for K and R are magnified to show the distribution of the APESS scores of the mutagenesis sequences in more detail. (**B**) To predict mutations with high infectivity using APESS, mutagenesis was performed using Wuhan-Hu-1, Alpha, Beta, Delta, BA.1, BA.2, BA.2.75, BA.4/BA.5, XBB, and BQ.1 as the backbone. The presence of these amino acid substitutions was verified using the viral sequence data from GISAID. For each lineage, the amino acid substitutions resulted in 280 mutagenic sequences. Thirty sequences with the highest APESS and pDockQ scores are displayed. N460R and S469R have not been observed naturally, whereas

*Figure 5 continued on next page*

*Figure 5 continued*

N439R, S459R, N437R, Y501R, S438R, and S494R have been observed in ten people or less. (**C**) For mutations occurring in lineages and mutations evaluated through APESS, AI learning models (Random Forest, LightGBM, XGBoost, Ensemble, and deep learning) were used to investigate the probability. For N460K, there was a ninefold increase in the probability of XBB compared to prior Omicron lineages. Q493R is not found in XBB but still has a high probability of occurrence. (**D**) The effects of N437 and N460 amino acid substitutions on viral infection were evaluated in vitro using luciferase and viral entry assays. There was a significant increase in the Relative Light Units and viral entry percentage for N437R, and vice versa for N460K.

The online version of this article includes the following figure supplement(s) for figure 5:

**Figure supplement 1.** Prediction results of notable mutations in SARS-CoV-2 using ML/DL.

**Figure supplement 2.** Protein structure evaluation of D467.

**Figure supplement 3.** Binding affinity measurement of N460K and Q493R receptor binding domain (RBD) variants.

**Figure supplement 4.** Flowchart for predicting mutation occurrences using artificial intelligence.

## Discussion

In this study, we aimed to comprehensively analyze the viral sequence and protein structure of SARS-CoV-2 and investigated its association with epidemiological data. Specifically, we analyzed the impact of these factors on infection and predicted the occurrence of new mutations. Our approach involved a multistep analysis. First, we identified specific amino acid substitutions within the viral genome, focusing on their potential impact on SARS-CoV-2 protein structure. Second, we explored the impact of these structural changes on the interaction between SARS-CoV-2 and the host. Third, we assessed the potential effects of these changes on virus infectivity. This systematic analysis allowed us to gain useful insights into the behavior and evolution of the virus.

Hydrophobic protein structure plays an important role in protein stability and folding (*Pace et al., 2011*; *Islam et al., 2019*). This affects structural changes from hydrophilic to hydrophobic or positively charged residues.

In our investigation of various viruses, Flaviviruses, including Zika virus, Japanese encephalitis virus, and Dengue virus, are enveloped viruses that use envelope proteins to infiltrate the host. Infection is facilitated by E-protein folding, which causes the virus to fuse with its host (*Hu et al., 2021*). Recently, SARS-CoV-2 treatments targeted the folding of the nonstructural protein NSP3 of the virus (*Bergasa-Caceres and Rabitz, 2020*). Research on avian coronaviruses has shown that the introduction of a hydrophobic domain into the infectious bronchitis virus E protein affects the infectivity of the virus (*Ruch and Machamer, 2011*). Amino acid substitutions such as L411F, F472S, D510S, and I529T have been reported in MERS-CoV (*Wong et al., 2021*; *Kleine-Weber et al., 2019*). In addition to D510S, consecutive hydrophilic amino acids were observed in MERS-CoV, which we believe contributed to its low infectivity (*Figure 1—figure supplement 4*).

Our analysis primarily focused on the RBM (amino acids 437–508) region of the spike protein of SARS-CoV-2 that directly interacts with the ACE2 receptor, allowing the virus to infiltrate host cells and discover mutations in VOCs and VUMs. In the Alpha, Beta, and Delta lineages, specific amino acid substitutions occurred in various locations. For Omicron, less diversity was observed for amino acid substitutions (*Figure 2B*).

Amino acid substitutions became fixed as SARS-CoV-2 lineages progressed. Examination of epidemiological data revealed four distinct periods. During the first and third periods, there was less diversity in amino acid substitutions, leading to a decrease in both infections and deaths (*Figure 3A*). In contrast, during the second period, there was more diversity in amino acid substitutions for the sublineages, leading to an increased number of cases and deaths. Thus, we established an association between the fixation of SARS-CoV-2 amino acid substitutions and infectivity.

We made discoveries on specific amino acid substitutions at positions. The N437R mutation led to increased viral infectivity and received a high APESS score but was barely observed in patients. We detected only three instances of N437R amino acid substitutions, all with the AAT codon, where two or more codon positions must undergo alterations. Such changes were only in V445P, a mutation found in XBB and its sublineages, but it occurred at a miniscule rate of 0.14%. Although not yet prominent, N437R is expected to be associated with high infectivity and remains a primary candidate for close monitoring in the future.

Frequency prediction of N460K through machine learning showed a ninefold increase compared to BA.1. And the mutation showed decreased viral entry in vitro and low binding affinity in silico. The

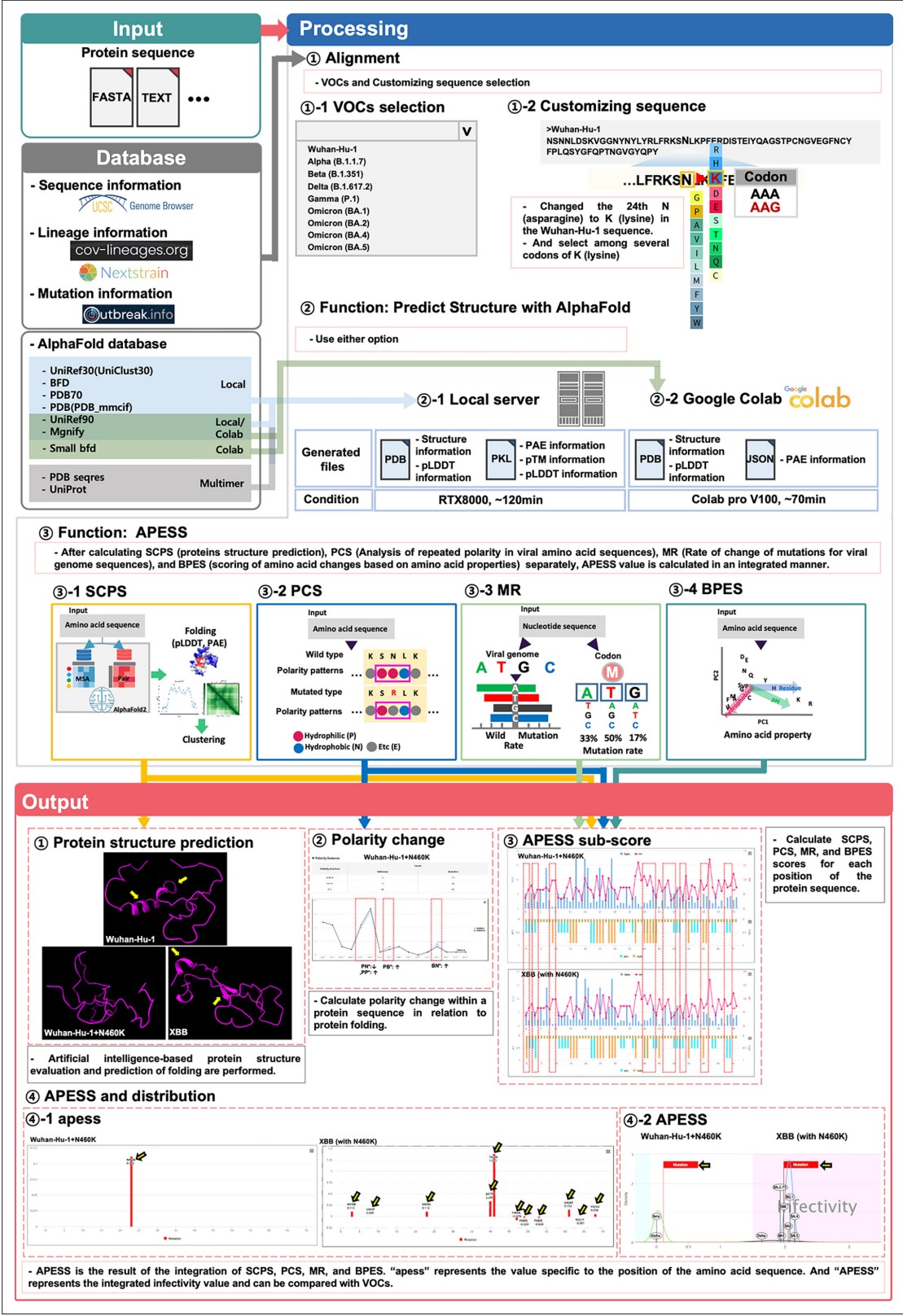

**Figure 6.** Prediction of potential SARS-CoV-2 mutations through integrated evaluation and prediction. (Input). This figure consists of three steps: 'Input,' 'Processing,' and 'Output.' Users can select a custom sequence from the entire SARS-CoV-2 sequence, choose variants of concerns (VOCs), or create customized sequences for analysis. Depending on the user's system environment, analysis can be done through local prediction (server) or Google Colab. (Processing) Three types of analyses are performed. First, the protein 3D structure prediction is analyzed. This includes protein 3D structure,

*Figure 6 continued on next page*

*Figure 6 continued*

predicted local distance difference test (pLDDT), and predicted aligned error (PAE). Second, the infectivity is evaluated using APESS (2.12). For each position, the structural difference graph for biochemical properties eigen score (BPES), mutation rate (MR), polarity change score (PCS), and sub-clustering of protein structure (SCPS) is visualized. The APESS distribution is visualized for known VOCs and created variants. Third, polarity changes are visualized in sequences. (Output) Four results comparing Wuhan-Hu-1+N460 K and XBB (with N460K) are output and visualized. First, through protein structure prediction, secondary structures can be confirmed in XBB (with N460K) compared to Wuhan-Hu-1+N460 K (Yellow arrow). Second, the comparison of polarity changes through the mutation of Wuhan-Hu-1+N460 K (red dotted line) is done. Third, in XBB (with N460K), which has more mutations than Wuhan-Hu-1+N460 K, the difference in values at each position in the protein sequence of SCPS, PCS, MR, and BPES is displayed (red dotted line). Fourth, the distribution of APESS, which represents the comprehensive value of SCPS, PCS, MR, and BPES, is shown. 'apess' indicates the score for each position in the customized protein sequence. In the case of XBB (with N460K), which has more mutations than Wuhan-Hu-1+N460 K, an apess score distribution of XBB (with N460K) has values from –0.079–1.385 is shown (Yellow arrow). X-axis presents the position in RBM (72aa) and Y-axis presents APESS score, respectively (④–1). APESS is the summed value of each position and can evaluate infectivity. A region including XBB (with N460K) shows infectivity due to many mutations, and also shows an increase in APESS score due to the N460K mutation. X-axis presents APESS score and Y-axis presents a density, respectively (④–2). AIVE comprehensively evaluates viral infectivity, protein structure, amino acid substitutions, and polarity changes in preexisting and potential SARS-CoV-2 sequences.

The online version of this article includes the following figure supplement(s) for figure 6:

**Figure supplement 1.** The user interface user experience (UI/UX) of the AIVE system.

**Figure supplement 2.** Frequencies of amino acid substitutions of major variants within clades.

N460K mutation affects docking but combination with other mutations could increase infectivity. Ito et al. confirmed that N460K mutation forms a hydrogen bond to N-linked glycan on N90 of human ACE2 (*Ito et al., 2023*).

The D467 position plays a key role in a salt-bridge interaction within the Delta variant (*Baral et al., 2021*). Due to this and mutations at the position being extremely rare suggest that D467 contributes to the structural stability of the virus. Specifically, amino acid substitutions D467P and D467I disrupt the salt bridge formed by R457 and R454 (*Figure 2—figure supplement 1*). Salt bridges play an important role in thermostabilization (*Ban et al., 2019*) and represent electrostatic interactions between positively charged and negatively charged residues. Alterations in positively charged residues can disrupt these salt-bridge interactions with negatively charged residues of SARS-CoV-2. Overall, mutations at this site impair the ability of SARS-CoV-2 to effectively infect the host. In silico results showed amino acid substitutions at D467 caused the alpha helix to present in the wild-type to change into a linear structure (*Figure 2—figure supplement 3*).

XBB predicted by ML exhibited a decrease in frequency, displaying a low slope and signifying slow disappearance (*Figure 5C*). Protein measurement proved to be difficult in vitro. Notably, there was an approximately twofold increase in binding affinity against the wild-type in SPR analysis. For the Q493R mutation itself, the docking is low but combined with other mutations, it showed increased docking. We believe that the mutation itself negatively affects the spike protein structure but is important in binding with the host overall. Recent structural research which identified RBD-ACE2 complex structures of BQ1.1 (N460K) and B1.1.529 (Q493R) supports our results (*Han et al., 2022*).

Our research shows that consecutive hydrophobic amino acids in the RBM region and specific amino acid substitutions affect not only infection but also protein structure. We assessed the infectivity of SARS-CoV-2 lineages using protein structure prediction. We observed that the emergence of various mutations corresponded to changes in binding affinity. As SARS-CoV-2 lineages progressed to Omicron, their binding affinity became stronger, resulting in increased infectivity, notably in the Delta and Omicron variants. Conversely, the Beta variant showed high binding affinity but not high infectivity. Our assumption is that for consecutive amino acids and amino acid substitutions, lysine (K) and arginine (R) have less weight; therefore, Beta is unlikely to develop into a significant variant. We speculate that the Beta variant emerged early in the pandemic, potentially limiting its ability to infect a large population, while widespread vaccination efforts may have contributed to decreased infectivity.

During the evolutionary process of SARS-CoV-2, the increase in infectivity and severity of variants from Alpha to Delta was attributed to abnormalities in the molecular signaling systems of the host. In contrast, Omicron showed lower severity but increased infectivity. We believe this is because of structural changes that allow the variant to bind well with the ACE2 receptor (*Figure 3*, *Figure 3—figure supplement 2*). In addition, epidemiological data reveals a decrease in deaths, the reproduction rate is maintained, and Omicron has high docking which confirms the high infectivity. We created

mutagenesis sequences created from the backbone of Wuhan-Hu-1 and VOCs with amino acid substitutions to lysine (K) and arginine (R). Sequences from the backbone of Omicron XBB were predicted to have high infectivity and still liable to occur. We can therefore expect consistent infections caused by SARS-CoV-2 going forward.

We comprehensively evaluated genomic and epidemiological data. This multifaceted approach, along with the usage of APESS, machine learning, in vitro assays, and AIVE ensured a more comprehensive understanding of SARS-CoV-2 behavior and evolution. In addition, by applying GMM to the APESS score of a new sample, we can predict its membership within a specific component (*Figure 4A*). This prediction allows us to make informed conjectures about its potential strains and infectivity.

Based on our findings, we predicted approximately 440 mutations with a probability of occurring in the near future. One year after generating the predicted datasets, BA.2.86, EG.5.1, and HK.3 were included in the datasets and were reported to the highest frequencies with 43%, 28%, and 12% at Nextstrain (https://nextstrain.org/ncov/gisaid/global/6m; December 21, 2023). APESS values of them also were high at 2.051, 1.982, and 1.986, respectively. These mutations were provided in AIVE and can be considered as important targets to prevent a new wave of infections.

We created AIVE to feature our findings and analyses on an online platform. AIVE can be easily utilized without expert knowledge of its algorithm and its analysis features can be used without GPU setup or the need for software library installations. Furthermore, AIVE offers a user-friendly interface with the flexibility to input and append experiment data for analysis. Additionally, ongoing and past analyses can be accessed after processing, providing many options for management. The time required for analysis in the local server or Google Colab can be verified. The local server run on RTX8000 takes 50 min, offers a large database, and is free to use (*Figure 6*). After completing the job, the user can examine and manage the analysis results permanently with their account generated on AIVE.

Prior research of SARS-CoV-2 included analysis of epidemiological data, molecular work, and AI applications. We adopted a comprehensive approach, utilizing real-world data and multi-faceted validation. Our research reveals that SARS-CoV-2 increased in infectivity over time, illuminating significant trends in viral infections. Our discoveries, evaluation model, and the AIVE platform will serve as the foundation for preparedness against further developments in future pandemics.

## Materials and methods
### Data collection
We used the World Health Organization tracking of SARS-CoV-2 variants to verify the currently circulating VOCs. In the tracking SARS-CoV-2 variants page, variants are divided into currently circulating VOCs, currently circulating variants of interest (VOIs), and currently circulating VUMs. In our study, the primary currently circulating VOCs were Omicron, including BA.1, BA.2, BA.3, BA.4, and BA.5 (https://www.who.int/activities/tracking-SARS-CoV-2-variants).

To obtain the precise lineage and mutation data, cov-lineages.org and outbreak.info databases were used. For the sublineage analysis of the viral sequences collected through GISAID, we used accurately defined sublineage information from the outbreak.info platform.

Using WHO VOCs as standards, Alpha (B.1.1.7), Beta (B.1.351), Delta (B.1.617.2), and Omicron (BA.1, BA.2, and BA.4/BA.5) were selected as the main lineages. From WHO's currently circulating VUMs as of 17 May 2023, we included Omicron BA.2.75 nicknamed 'Centaurus,' BQ.1, and XBB.

We downloaded the mutation data of the currently reported 7,335,614 viral sequences from GISAID and utilized the data for the following purposes: first, the epidemiological data were categorized based on the patient status column: symptomatic or asymptomatic, and severity: mild or severe (including hospitalized and deceased). Second, for VOCs, including sublineages, we categorized lineage, date, and mutations in the RBM region. Third, to utilize the learning data, we categorized clades, lineages, and mutations in the RBM region. Our epidemiological data on COVID-19 used our world's data (OWID, https://ourworldindata.org/). For confirmed cases and deaths, we used 7 d rolling average worldwide data. For vaccine doses, cumulative vaccine doses were used.

## Detection of amino acid features from SARS-CoV-2 sequences

We defined four properties: hydrophilicity (polar, P), hydrophobicity (non-polar, N), alkalinity (basic, B), and acidity (A). We divided two consecutive positions of amino acids with these properties into NN*, NP*, NA*, NB*, PN*, PP*, PA*, PB*, AN*, AP*, AA*, AB*, BN*, BP*, BA*, and BB*, with the most common combination being NN*, NP*, PN*, and PP*.

We verified consecutive amino acids in the SARS-CoV-2 viral sequence spike protein. First, we categorized alanine (A), valine (V), leucine (L), glycine (G), isoleucine (I), methionine (M), tryptophan (W), phenylalanine (F), and proline (P) as hydrophobic amino acids. Second, we categorized serine (S), cysteine (C), asparagine (N), glutamine (Q), threonine (T), and tyrosine (Y) as hydrophilic amino acids. Third, lysine (K), arginine (R), and histidine (H) were categorized as alkaline amino acids (B: positively charged). Finally, aspartic acid (D) and glutamic acid were categorized as acidic amino acids.

We performed the following analysis to observe the polarity changes due to specific amino acid substitutions. Based on polarity and considering the possibility of mutations occurring, we classified each adjacent amino acid for each mutation and divided them into 64 patterns of three amino acids grouped together. We determined the pattern changes caused by the mutations. The analysis methods are available publicly on the web-based platform AIVE (https://xn--aive-186a.org/) and GitHub (https://github.com/Honglab-Research/AIVE, copy archived at *Honglab-Research, 2023*).

## Protein 3D structure evaluation

We utilized Alphafold2 to measure protein structure folding and used pDockQ and HADDock to measure the binding affinity between SARS-CoV-2 and the ACE2 receptor (*Khan and Ranganathan, 2010*). We used AlphaFold2 to predict changes in the 3D structure of the RBM at amino acid positions 437–508 in the SARS-CoV-2 viral sequence. We performed predictions for Wuhan-Hu-1, Alpha, Beta, Delta, and Omicron (BA.1, BA.2, BA2.75, BA.4, BA.5, XBB, and BQ.1).

Through UniProt (P0DTC2 · SPIKE_SARS2), we used the viral sequence in the RBM region 'NSNN LDSKVGGNYNYLYRLFRKSNLKPFERDISTEIYQAGSTPCNGVEGFNCYFPLQSYGFQPTNGVGYQPY.' We performed an analysis using mutation data per lineage (https://www.uniprot.org/uniprotkb/P0DTC2/entry).

We used AlphaFold2 and through structure predictions, we observed structural similarity, distance, and folding between amino acids. The predicted local distance difference test (pLDDT), Predicted Aligned Error (PAE), and AlphaFold2 results were used for analysis. Additionally, protein data bank (PDB) files and another AlphaFold2 result, were used to visualize the 3D structures using ChimeraX (*Pettersen et al., 2021*). We measured the binding affinity between the human ACE2 receptor and the SARS-CoV-2 RBM region. We obtained the human ACE2 receptor sequence from UniProt (UniProt: Q9BYF1).

To increase the confidence of binding affinity analysis, first, we used pDockQ to score binding affinity. Second, we used HADDOCK 2.2 for analysis and comprehensively evaluated the HADDOCK score, root mean square deviation (RMSD), van der Waals energy, electrostatic energy, and desolvation energy (*Figure 3—figure supplement 2*). AIVE can predict the structural features of SARS-CoV-2 in different regions and the binding affinity between the human ACE2 receptor and the RBM region of SARS-CoV-2.

## Analysis of the association between epidemiology and mutations

From 7,335,614 viral sequences sourced from GISAID, we identified mutations commonly discovered in VOCs and VUMs: L440K, K444T, L452R, N460K, S477N, T478K, E484A, F486V, Q498R, and Y501H. We measured the odds ratios for mutations commonly discovered in VOCs and VUMs: L440K, K444T, L452R, N460K, S477N, T478K, E484A, F486V, Q498R, and Y501H, and epidemiological indicators (symptomatic/asymptomatic and mild/severe). These analyses were performed using the traditional statistical method of logistic regression.

In the patient status column of the data file, we used asymptomatic and symptomatic rows. Deceased, hospitalized, and severe patients were categorized as severe, while outpatients, not hospitalized, and mild were categorized as mild. R version 4.1.1 and python 3.9.13 were used in our analysis.

# Evaluation of SARS-CoV-2 mutation infectivity through evaluation modeling

We developed an evaluation model to comprehensively evaluate SARS-CoV-2 at the RNA, amino acid, and protein structure levels. SCPS, PCS, MR, and BPES were designed and calculated for each variant.

## SCPS

To better understand the molecular structure of SARS-CoV-2 lineages, we used the PAE, pLDDT, pdb, and hydrogen bonds that caused abnormal physical and chemical properties of the compounds to proceed with clustering.

Using clustering, the SARS-CoV-2 lineages were divided into six clusters, and the main variants were mostly observed in the first and third clusters. For the cluster distribution for each lineage, 'cluster 2' showed the largest difference in folding compared to the Wuhan-Hu-1 variant, and the lowest difference in folding compared to the Delta and Omicron (BA.4/BA.5) variants. Meanwhile, 'cluster 4' contained the lineages with the most extreme folding showed a relaxation of folding in the main variants.

Based on this analysis, protein structure prediction-based clustering was defined as follows:

$$\text{SCPS}_i = \begin{cases} 0.9 \; if \, SCPS_i \in \{\text{T2, T4}\} \\ 0.1 \; if \, SCPS_i \in \{\text{T1, T3, T5, T6}\} \end{cases}$$

where T1 is cluster 1 (437–449), T2 is cluster 2 (450-460), T3 is cluster 3 (461-471), T4 is cluster 4 (472-479, 501-508), T5 is cluster 5 (480-489), and T6 is cluster 6 (490-500), which are positioned in the RBM of the RBD.

Omicron and Delta variants take up 88.3% of the 7,335,614 viral sequences from GISAID. Therefore, we applied a weight of 0.9 to the T2 and T4 clusters and 0.1 to the remaining clusters.

## PCS

To calculate the amino acid substitutions in the RBM sequence, we counted each lineage: Wuhan-Hu-1, Alpha, Beta, Delta, and Omicron (BA.1, BA.2, BA2.75, BA.4, BA.5, BQ.1, and XBB). For the counting method, we started from the beginning of the RBM, sliding three positions after each count until the final 72$^{nd}$ position was reached.

$$\text{PCS}_i(w, m) = \begin{cases} 0.9 \; if \, PCS_i(w_i) \in \; PP^* \Lambda PCS_i(m_i) \notin PP^* \\ 0.5 \; if \, PCS_i(w_i) \in \; PP^* \Lambda PCS_i(m_i) \notin PP^* \\ 0.1 \; if \, otherwise \end{cases}$$

$$PP^* = \left[ (a_1, a_2, a_3) | a_1, a_2 \in P, a_3 \notin P \right], \text{P is hydrophilic set and } a_1 \text{ is an amino acid}$$

Three consecutive amino acids with two hydrophilic amino acids in Wuhan-Hu-1 changes due to mutation (0.9), when the inverse occurs (0.5), and for mutations unrelated to consecutive hydrophilic amino acids (0.1).

## MR

Based on the base trans rate and codon compositional bias, we investigated the ratio of A, T, G, and C in SARS-CoV-2 spike protein nucleotides and calculated the rate of change of bases depending on the mutations occurring. On average, the SARS-CoV-2 virus comprises 30.2% A, 19.9% G, 32.4% U, and 17.6% C. For 17 lineages {Alpha (B.1.1.7), Beta (B.1.351), Gamma (P.1), Delta (B.1.617.2), Zeta (P.2), Epsilon (B.1.427), Epsilon (B.1.429), Eta (B.1.525), Lambda (C.37), Mu (B.1.621), Omicron (BA.1), Omicron (BA.2), Omicron (BA.4), Omicron (BA.5), Omicron (BA.2.75), Omicron (BQ.1), and Omicron (XBB)}, we checked the differences between the codon positions where mutations occurred. We set the first codon at 28%, second codon at 49%, and the third codon at 22% to create an amino acid mutation property equation based on biological properties.

$$MR_i(w, m) = \frac{mr(w_i) - mr(m_i)}{mr(w_i)} \quad mr(x) = \prod_{j=1}^{3} N_j(x) \times C_j(x)$$

For each codon position, $MR_i(w,m)$ calculates the frequency of mutation (m) against the reference (w). $mr(x)$ evaluates the frequency change for each nucleotide in a codon. $N_j(x)$ is the frequency for j-the nucleotide. $C_j(x)$ is the frequency for j-the codon position: $C_1$ (first codon 28%), $C_2$ (second codon 49%), and $C_3$ (third codon 22%).

The total mutation rate interval is [0.0072, 0.0201] for 64 codons. 16 amino acids (G, E, A, V, R, K, N, T, I, Q, H, L, W, C, Y, and S) are amino acid substitutions that occur in SARS-CoV-2. The mutation rate interval for these amino acid substitutions is [0.0115, 0.0137].

## BPES

We used amino acid properties such as residue, pH, and hydrophobic properties using principal component analysis (PCA). From VOCs, we analyzed the 72 amino acids included in the RBM region. Regarding amino acid properties, amino acids with residues C, D, E, K, R, and Y were in the order R:12.48, K:10.53, and Y:10.07. pH was in the order of R:10.76, K:9.74, and H:7.59. The hydrophobic amino acids were in the order of R: –4.5 and K: –3.9, where K and R showed similarly high scores. K and R received high scores for all the three amino acid properties (*Supplementary file 1x*). Based on the three biochemical properties of amino acids (residue, pH, and hydrophobicity) and the PCA clustering, we defined the eigenscore as follows.

$$BPES_i(w, m) = \frac{bpes(w_i) - bpes(m_i)}{bpes(w_i)}$$
$$bpes(x) = \|R(x)\| \frac{R(x) \cdot P(x)}{\|R\| \cdot \|p(x)\|} + \|P(x)\| + \|H(x)\| \frac{H(x) \cdot P(x)}{\|H\| \cdot \|p(x)\|}$$

For each amino acid property, $BPES_i(w,m)$ calculates the value of the mutation (m) against the reference (w). For $bpes(x)$, R is the eigen vector of a residue score, H is the eigen vector of a hydrophobic score, P is the eigen vector of the pH score, $\| \|$ is a norm of the vector, x is the amino acid, and is the dot product.

APESS

$$APESS(w, m) = \sum_{i=1}^{n} SCPS_i(w) \times PCS_i(w, m) \times MR_i(w, m) \times BPES_i(w, m)$$

APESS is the comprehensive value of SCPS, PCS, MR, and BPES. n is the number of positions, w means wild, and m means mutated.

The Kernel Density Estimator (kdeplot) from Python's seaborn package was used to create a distribution function for APESS scores calculated across randomly sampled 30,000 SARS-CoV-2 sublineages sourced from GISAID. For all 7,335,614 sequences, we also calculated the scores and studied the distribution. This distribution function revealed that the SARS-CoV-2 lineages could be differentiated and characterized based on their APESS scores (*Figure 5A*). Based on this observation, we applied GMM to identify unique components of APESS score. We divided the range of the APESS score into two regions: the lower region for scores less than or equal to 3.2 and the upper region for scores greater than 3.2. To select the optimal number of components for GMM in each region, we used Bayesian Information Criterion (BIC), resulting in 4 and 5 as the optimal number of components in the lower and upper regions, respectively. We fit GMM for each region to construct prediction models with the corresponding optimal number of components.

## Analysis of gene regulation from SARS-CoV-2

We analyzed the differences in gene expression for various SARS-CoV-2 lineages when they infect the host. We downloaded GSE235262 from the Gene Expression Omnibus (GEO) to obtain data on human

gene expression. Comparisons were made between controls (uninfected people), Alpha (B.1.1.7), Delta (B.1.617.2), Omicron (B.1.529), Omicron (BA.2), and Omicron (BA.4/BA.5 and its sublineages). Only the genes that showed significant expression levels (log2 (TPM +1)) were selected. We used enrichR (https://currentprotocols.onlinelibrary.wiley.com/doi/10.1002/cpz1.90) for the enrichment analysis of these selected genes then visualized the comparison between Delta and Omicron (BA.4/BA.5).

## SARS-CoV-2 variant prediction and machine learning model

The model predicted the probability of mutations in the RBM region of Alpha, Beta, Delta, and Omicron (BA.1, BA.2, BA.2.75, BA.4, BA.5, BQ.1, and XBB) variants and the 24 mutagenesis variants generated using machine learning. Eighteen variants were used for probability prediction. These were (N440K, V445P, G446S, N460K, S477N, T478K, E484A, F486P, F490S, Q498R, N501Y, Y505H, L452R, E484K, Q493R, G496S, F486V, and K444T). Additionally, a total of 24 VOCs (S477R, S477K, N439R, Y501R, N437R, S438R, S459R, S469R, S494R, T470R, T500R, Q493K, Q493R, T470K, T500K, S469K, S494K, Y501K, N437K, N439K, N460R, N460K, S438K, and S459K) excluding overlapping variants and variants lacking expression in data were also used for probability prediction (*Figure 6—figure supplement 2*).

The prediction was carried out using data collected from 2019-12-23 to 2023-05-17 (1242 d), and then with six different types of data, excluding specific clades:

- The first group consists of clades before 21 M (Omicron B.1.1.529) including 19 A, 19B, 20 A, 20B, 20 C, 20D, 20E, 20 F, 20 G, 20 H, 20I, 20 J, 21 A, 21B, 21 C, 21D, 21E, 21 F, 21 G, 21 H, 21I, and 21 J.
- The second group consisted of clades before 21 K (Omicron BA.1) and 21 L (Omicron ~BA.2), including the prior group and the addition of 21 M, 21 K, and 21 L.
- The third group consisted of clades before 22D (Omicron BA.2.75) and 22 C (Omicron BA.2.12.1), including the prior group and the addition of 22 C and 22D.
- The fourth group consists of clades before 22 A (Omicron BA.4) and 22B (Omicron, BA.5), including the prior group and the addition of 22 A and 22B.
- The fifth group consisted of clades before 22E (Omicron BQ.1), including the prior group and the addition of 22E.
- The sixth group consisted of clades before 22 F (Omicron XBB), 23 A (Omicron XBB 1.5), and 23 B (Omicron XBB), including the first group and the addition of 22 F, 23 A, and 23 B.

Machine learning determined the clade information and presence of mutations in 24 variants. The possibility of each variant is extracted through learning. Machine learning was performed using a LightGBM, which is a highly efficient gradient boosting decision tree, XGBoost, which is a scalable tree boosting system, Random Forest, which is a random forest guided tour, and an ensemble model that combined the three previous models. To calculate the probability of mutation occurrences, our custom script and the following libraries were used in Python 3.9.13. Data processing employed sci-kit-learn for scaling, data splitting, and accuracy assessment. In the case of machine learning, LightGBM and XGBoost utilized the 'lightgbm' and 'xgboost' libraries, respectively, to build their models. Meanwhile, the Random Forest and ensemble models utilized sci-kit-learn's subfunctions to create their models. The deep learning model used TensorFlow and its lower-level library, Keras, for model development.

## The flow of data preprocessing and machine learning

We retrieved information from SARS-CoV-2 samples collected on GISAID from December 23, 2019, to May 17, 2023, spanning 1242 d. We filtered the data using various columns, including date, Nextstrain clade, species, substitution, AA substitution, deletion, and insertion, among others.

First, for species data, we extracted the data where the host was either 'homo' or '*Homo sapiens.*' Then, we removed the samples with date values that did not follow the 'YYYY-MM-DD' format. Based on the insertion, deletion, and AA substitution columns, we removed the samples where deletions, insertions, or nonsense mutations occurred. Following this, we removed columns other than date, clade, and lineage, and then eliminated samples with missing values (NaN).

For the extracted samples, we organized the amino acid mutation information from the AA substitution column for the 437–508 region of the S protein. We created columns indicating the presence (1)

or absence (0) of 40 target mutations. We removed mutations not found in the GISAID data, leaving the columns representing the presence or absence of 24 mutations (N440K, V445P, G446S, N460K, S477N, T478K, E484A, F486P, F490S, Q498R, N501Y, Y505H, L452R, E484K, Q493R, G496S, F486V, K444T, S494R, T500R, Q493K, T470K, N439K, and N460R).

To prepare the filtered data for training, we transformed the values for the dates into integers, representing the number of days that had passed since the initial collection date. We then normalized these values to fall within the range of [0–1]. Using one-hot encoding, we preprocessed the Nextstrain clade information. From the preprocessed data, we created six datasets, each based on different clade groups. Finally, we used the random state function to generate the training and testing datasets used in the learning process.

The training data is used for learning as the input for five different models created. The learning model then predicts the likelihood of the occurrence of 24 target mutations (*Figure 5—figure supplement 4*).

## Cell culture

HEK293T and 293T-hACE2 cells were maintained in Dulbecco's Modified Eagle Medium (DMEM, Welgene, South Korea) supplemented with 10% fetal bovine serum (FBS, Gibco) and 1% penicillin-streptomycin (Gibco) at 37°C in a humidified 5% $CO_2$. A HEK293T cell line stably expressing human ACE2, 293T-hACE2, was generated using a lentivirus-mediated gene transduction system under the antibiotic pressure of hygromycin. The identity of all cell lines was authenticated using short tandem repeat (STR) profiling. All cell lines were tested for mycoplasma contamination using PCR-based assays and confirmed to be mycoplasma-free.

## Molecular cloning and plasmid construction

The human codon-optimized full-length SARS-CoV-2 spike (Wuhan-Hu-1 strain) gene was obtained from Sino Biological Inc (Beijing, China). The last 19 amino acids of the SARS-CoV-2 spike were truncated to increase protein expression, and a flag tag was inserted at the C-terminal end. Spike gene mutation constructs, including D614G, N437R, N460K, D467P, D467I, and Q493R, were generated using a PlatinumTM SuperFi ll PCR Master Mix (Invitrogen, Waltham, MA, USA).

## Spike protein expression test

Viral spike genes were cloned into pcDNA3.1(+), and plasmid DNA was transfected into HEK293T cells using the PEI reagent (Sigma). After 48 hr, the cells were lysed with a 1% NP-40 lysis buffer (150 mM NaCl, 1% NP-40, 50 mM Tris-HCl), and the extracted proteins were analyzed by SDS-PAGE and immunoblotting. A mouse anti-FLAG antibody (Sigma) was used to detect spike protein expression.

## Pseudotyped virus and viral entry

The pseudotyped virus was generated from HEK293T cells (Invitrogen) by co-transfection with human immunodeficiency virus backbone plasmids expressing firefly luciferase as described previously. We used packaging plasmids (pLP1, pLP2, and pLP/VSV-G; all from Invitrogen) and pLVX-Luc-IRES-ZsGreen1_ Cat. 632187 (Luc stands for luciferase, and IRES stands for internal ribosomal entry site). For S protein pseudotyping, the full-length cDNA of the S gene (Sino Biological Inc) was cloned into pcDNA3 and used for transfection instead of pLP/VSV-G. A plasmid carrying the gene encoding the indicated mutation in the S protein was generated using a PlatinumTM SuperFi ll PCR Master Mix (Invitrogen, Waltham, MA, USA) based on the wild-type construct, and the point mutation was confirmed by sequencing. Viral supernatants were harvested 48 hr after transfection and normalized using the Lenti-X reverse transcription-quantitative PCR (qRT-PCR) titration kit _ Cat. 631235 according to the manufacturer's protocol. Infected 293T-hACE2 cells were lysed 48 hr after infection and the efficiency of viral entry was measured by comparing the luciferase activity of pseudotyped viruses bearing the wild-type or mutant S protein. A VSV-G-pseudotyped lentivirus was used as a positive control. Relative luciferase activity in the cell lysates was measured using a luciferase assay kit (Promega).

## Cloning, expression, and purification of ACE2, SARS-CoV-2 RBD, and RBD variants

The SARS-CoV-2 spike protein RBD, and its mutations (N460K and Q493R), along with the N-terminal peptidase domain of human ACE2 was expressed using the Bac-to-Bac baculovirus system (Invitrogen). The RBD (Residues Arg319-Phe541) and the N-terminal peptidase domain of human ACE2 (Residues Ser19-Asp615) with an N-terminal gp67 signal peptide were cloned upstream of the cleavable deca-histidine tag and FLAG tag. RBD variants were introduced into the wild-type RBD construct using PCR-based site-directed mutagenesis. The resulting constructs were expressed in Spodoptera frugiperda (Sf9) insect cells and secreted into the medium after being cultured at 27 °C for 72 hr.

Subsequently, ACE2, RBD, and its variants were isolated from the filtered supernatant using a HisTrap Excel column (Cytiva, Marlborough, MA, USA). The C-terminal tags were cleaved using an in-house-generated HRV3C protease. The protease and cleaved C-terminal tags were removed using Ni-NTA resin (Takara, 635662), and the proteins were further purified by size-exclusion chromatography on a HiLoad 26/600 Superdex 200 pg (Cytiva). Purification was performed in a buffer containing 20 mM Tris-HCl (pH 7.5), 150 mM NaCl, and 1 mM DTT. To maintain stability during storage, each protein sample was frozen at –80 °C and supplemented with 10% glycerol. For surface plasmon resonance (SPR), proteins were thawed from frozen storage immediately before the experiment.

## Binding affinity measurement

The SPR experiments were performed at room temperature using BiaCore T200 with a series of S CM5 sensor chips (Cytiva). The surfaces of the sample and reference flow cells were activated using a 1:1 mixture of NHS (N-hydroxysuccinimide) and EDC (3-(N,N-dimethylamino) propyl-N-ethylcarbodiimide). The reference flow cell was left blank. All the surfaces were blocked with ethanolamine (pH 8.0). HBS-EP + (Cytiva) was used as a running buffer.

For the binding affinity assay, the purified N-terminal domain of ACE2 was diluted in 10 mM sodium acetate buffer (pH 4.0) and immobilized on the chip at approximately 800 response units. The SARS-CoV-2 RBD and RBD mutations along the gradient were observed on the chip surface. After each cycle, the sensor surface was regenerated using 10 mM glycine-HCl (pH 2.5). The data were fitted to a 1:1 interaction steady-state binding model using BIAevaluation 3.1 software. Curve fitting was performed via nonlinear regression using a one-site-specific binding equation in GraphPad Prism version 8.4.0 (GraphPad Software, Boston, MA, USA).

## AIVE platform

The AIVE platform (https://ai-ve.org/) provides real-time protein structure predictions and APESS score calculations. AIVE reports the analysis results and mutation data from known sequences, randomly sampled SARS-CoV-2 sequences, or custom sequences provided by users. Based on the mutations in the SARS-CoV-2 lineages, Alphafold2 was utilized to analyze the protein structures. Using the Wuhan-Hu-1 sequence as a reference, we performed a comparative analysis of the main VOCs: Alpha, Beta, Delta, and Omicron. Reports are generated using visualization tools that provide Alphafold2 outputs such as pLDDT and PAE. Monomer folding and docking were visualized and scored. Differences caused by mutations are shown in the input amino acid sequences based on polarity and protein properties. APESS can predict the infectivity of sequences through the distribution of values obtained through the GMM prediction model and using previously known lineages. The results of the user's APESS sequence scores can be visualized. AIVE runs on a high-performance computing (HPC) system with 3 RTX8700 GPUs, 96 CPUs, and 256 GB of main memory (*Supplementary file 2*).

# Acknowledgements

We thank the Global Science experimental Data hub Center (GSDC) and the Korea Research Environment Open NETwork (KREONET) service for data computing and networks provided by the Korea Institute of Science and Technology Information (KISTI).

## Additional information

### Funding

| Funder | Grant reference number | Author |
| --- | --- | --- |
| National Research Foundation of Korea | NRF-2021M3H9A2097227 | Dongwan Hong |
| National Research Foundation of Korea | NRF-2022R1A2C3008162 | Dongwan Hong |
| National Research Foundation of Korea | RS-2023-00220840 | Dongwan Hong |
| Korea Health Industry Development Institute | RS-2023-00265923 | Dongwan Hong |
| National Research Foundation of Korea | NRF: 2021M3A9I2080490 | Nam-Hyuk Cho |
| Korea Health Industry Development Institute | RS-2024-00406488 | Jongkeun Park |

The funders had no role in study design, data collection and interpretation, or the decision to submit the work for publication.

### Author contributions

Jongkeun Park, Conceptualization, Data curation, Software, Formal analysis, Supervision, Validation, Investigation, Visualization, Methodology, Writing – original draft, Project administration, Writing – review and editing; WonJong Choi, Do Young Seong, Conceptualization, Data curation, Software, Formal analysis, Validation, Investigation, Visualization, Methodology, Writing – original draft, Writing – review and editing; Seungpil Jeong, Ju Young Lee, Hyo Jeong Park, Dae Sun Chung, Data curation, Formal analysis; Kijong Yi, Data curation; Uijin Kim, Ga-Yeon Yoon, Hyeran Kim, Taehoon Kim, Sooyeon Ko, Hyun-Soo Cho, Nam-Hyuk Cho, Validation; Eun Jeong Min, Formal analysis, Validation; Dongwan Hong, Conceptualization, Resources, Data curation, Software, Formal analysis, Supervision, Funding acquisition, Validation, Investigation, Visualization, Methodology, Writing – original draft, Project administration, Writing – review and editing

### Author ORCIDs

Hyun-Soo Cho ⓘ https://orcid.org/0000-0003-4067-4715
Dongwan Hong ⓘ https://orcid.org/0000-0002-7816-1299

Reviewer #1 (Public Review): https://doi.org/10.7554/eLife.99833.3.sa1
Reviewer #2 (Public review): https://doi.org/10.7554/eLife.99833.3.sa2
Author response https://doi.org/10.7554/eLife.99833.3.sa3

## Additional files

### Supplementary files

Supplementary file 1. Supplementary tables. (a) The number of polarity changes in the receptor binding motif (RBM). (b) The number of polarity changes in the spike protein. (c) Configuration of polarity changes caused by variants of concern (VOCs) in the RBM. (d) The number of amino acids in VOCs. (e) The number of transitions and transversions in the spike protein from viral sequences. (f) Epidemiological information (cases and deaths per million population) from Our World in Data (OWID). (g) Amino acid substitution of sublineages. (h) Symptomatic information (symptomatic and asymptomatic) from Global Initiative on Sharing All Influenza Data (GISAID). (i) Severity information (severe and mild) from GISAID. (j) Gene expression data between the Delta variant and Omicron (BA.4/5) variant from Gene Expression Omnibus (GEO) dataset. (k) Binding affinity between proteins in protein 3D structure using pDockQ. (i) Evaluation of hydrogen bonds (H-bond) through protein 3D structure prediction of SARS-CoV-2. (m) Binding affinity between proteins in protein 3D structure using HADDOCK. (n) The number of mutations per codon in the RBM. (o) The number of mutations

per codon in the receptor binding domain (RBD). (p) The number of mutations per codon in the spike protein. (q) Amino acid property eigen selection score (APESS) score for each mutation with VOCs as the backbone. (r) Prediction of mutation probability using lightGBM. (s) Prediction of mutation probability using XGBoost. (t) Prediction of mutation probability using Random Forest. (u) Prediction of mutation probability using ensemble. (v) Prediction of mutation probability using deep learning. (w) The number of samples by clade in global SARS-CoV-2 analysis data. (x) Amino acid property values utilizing principal component analysis (PCA) clustering.

MDAR checklist

Supplementary file 2. AIVE usage guide.

### Data availability

All data supporting the findings reported here are available in the paper and Supplementary Information. The raw and processed data, custom scripts, and codes used in this study are deposited in the GitHub repository (https://github.com/Honglab-Research/AIVE, copy archived at *Honglab-Research, 2023*), https://github.com/Honglab-Research/AIVE-prediction (copy archived at *Honglab-Research, 2024*) and AIVE (https://ai-ve.org).

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
