## [Editor Report · eLife Assessment]

The study provides **valuable** insight into the biological significance of SARS-CoV-2 by using a series of computational analyses of viral proteins. While the evidence is **solid**, the reviewers noted a lack of clarity about the objectives of the analyses. While impactful for the field, the manuscript would benefit from improved presentation.

---

## [Referee Report · Reviewer #1 (Public Review)]

Summary:

Park et al. conducted various analyses attempting to elucidate the biological significance of SARS-CoV-2 mutations. However, the study lacks a clear objective. The specific goals of the analyses in each subsection are unclear, as is how the results from these subsections are interconnected. Compiling results from unrelated analyses into a single paper can be confusing for readers. Clarifying the objective and narrowing down the topics would make the paper's purpose clearer.

The logic of the study is also unclear. For instance, the authors developed an evaluation score, APESS, for analyzing viral sequences. Although they state that the APESS score correlates with viral infectivity, there is no explanation in the results section about why this is the case.

In summary, I recommend reconsidering the structure of the paper.

---

## [Referee Report · Reviewer #2 (Public review)]

Summary:

The authors have developed a machine learning tool AIVE to predict the infectivity of SARS-CoV-2 variants and also a scoring metric to measure infectivity. A large number of virus sequences were used with very detailed analysis that incorporates hydrophoic, hydrophiclic, acid and alkaline characteristics. The protein structures were also considered to measure infectivity and search for core mutations. The study especially focused on the S protein of SARS-CoV-2. The contents of this study would be of interest to many researchers related to this area and the web-service would be helpful to easily analyze such data without indepth bioinformatics expertise.

Strengths:

- Analysis on large scale data

- Experimental validation on a partial set of searched mutations

- A user-friendly web-based analysis platform that is made public

Weaknesses:

- Complexity of the research

Comments on revisions:

The authors have addressed all my comments and is much more readable.

---

## [Author Response]

The following is the authors’ response to the original reviews.

**Reviewer #1 (Public review):**
Summary:Park et al. conducted various analyses attempting to elucidate the biological significance of SARS-CoV-2 mutations. However, the study lacks a clear objective. The specific goals of the analyses in each subsection are unclear, as is how the results from these subsections are interconnected. Compiling results from unrelated analyses into a single paper can be confusing for readers. Clarifying the objective and narrowing down the topics would make the paper's purpose clearer.The logic of the study is also unclear. For instance, the authors developed an evaluation score, APESS, for analyzing viral sequences. Although they state that the APESS score correlates with viral infectivity, there is no explanation in the results section about why this is the case.The structure of the paper should be reconsidered.

Thank you for your feedback. We have heeded the input that the study lacks a clear objective and made sure that the overall goal of the study is reflected in the Abstract, Results, and Discussion.

We have made sure that the specific goals in each subsection are clearer in the Results section that better explain the goals of those sections and elaborated on how the components of our study connect to each other. We have addressed these in more detail in the ‘Recommendations for the authors’ section.

Thank you for the feedback on APESS, our evaluation model. APESS was created based on virus properties that we discovered of SARS-CoV-2 in our study. When applying our evaluation model, high APESS scores indicated high infectivity. APESS is calculated from a comprehensive evaluation of SARS-CoV-2 at the nucleotide, amino acid, and protein structure levels.

The detailed explanations and exact calculations of APESS are detailed in the Materials and Methods section in line 571 but we should have been more detailed in the Results section as well. We have made sure to properly indicate this in the Results section in line 284.

And overall, we have made edits to the manuscript that accurately explain our research by amending terms, restructuring arguments, and providing more clarity for the interconnectivity of the research.

**Reviewer #2 (Public review):**
Summary:The authors have developed a machine learning tool AIVE to predict the infectivity of SARS-CoV-2 variants and also a scoring metric to measure infectivity. A large number of virus sequences were used with a very detailed analysis that incorporates hydrophobic, hydrophilic, acid, and alkaline characteristics. The protein structures were also considered to measure infectivity and search for core mutations. The study especially focused on the S protein of SARS-CoV-2. The contents of this study would be of interest to many researchers related to this area and the web service would be helpful to easily analyze such data without in-depth bioinformatics expertise.Strengths:- Analysis of large-scale data.- Experimental validation on a partial set of searched mutations.- A user-friendly web-based analysis platform that is made public.Weaknesses:- Complexity of the research.

Thank you for your kind feedback. Our study explored a wide range of topics including biochemical properties, machine learning, and viral infectivity.

In presenting our research, we recognize that our comprehensive analysis may have slightly obscured the specific aims and overall objective of the study. We investigated properties in the viral sequences of SARS-CoV-2 and examined big data, clinical data, and expression data to elucidate their effect on viral infectivity. We then used evaluation modeling and in silico and in vitro validation.

We have clarified the aims of our research and improved upon the flow of the manuscript by adding sentences that outline the goals of our research in the appropriate sub sections of the Results and Discussion sections.

**Reviewer #1 (Recommendations for the authors):**
The abstract should clearly state the backgrounds, objectives, strategies, and findings of this study in an orderly manner.

Thank you for your feedback. We have restructured the Abstract to better reflect the goals and methods of our study. We start the Abstract by introducing the background of the study ‘An unprecedented amount of SARS-CoV-2 data has been accumulated compared with previous infectious diseases, enabling insights into its evolutionary process and more thorough analyses.’ in line 48. Then we more clearly stated the overall objectives of our research in line 50 as ‘This study investigates SARS-CoV-2 features as it evolves to evaluate its infectivity.’ Then, we clearly defined our specific discoveries in the virus, the purpose of our evaluation model, and how we validated our findings.

In the Introduction, the message of each paragraph is unclear. Please clearly state the objectives of the study and what was done to achieve these objectives.

Thank you for the feedback. We have updated the Introduction section to more clearly state the objectives of the study.

To increase clarity, we have moved ‘Furthermore, hydrophobic properties in the amino acid sequence affect protein folding. Coronavirus hydrophobicity has significant effects on amino acid properties and protein folding.’ to line 127.

In line 130, we rephrased the first sentence of the paragraph to ‘For these prior approaches to virus analysis and prediction, expertise with the relevant fields is required for a full understanding.’ to better establish the link between the background information and aims of the study. Then in line 134, we added ‘elucidate properties about the virus’ to clarify the aims of the study.

In line 141, we have improved the clarity of the sentence to better present the scope and objectives of the study.

The relationship between the sections in the Results is unclear. Clarify why each section is necessary and how they are interconnected.

We investigated properties in the viral sequences of SARS-CoV-2 that highlighted amino acid substitutions or changes in polarity (Figure 1). In VOCs, we noted trends or absences of amino acid substitutions at specific positions (Figure 2). We examined epidemiological and clinical data to determine the infectivity, severity, and symptomaticity of lineages. Looking at expression data and binding affinity further illuminated the effect of amino acid substitutions (Figure 3). We created APESS, an evaluation modeling, that is comprehensively calculated from the nucleotide, amino acid, and protein structure levels of the virus. Evaluation of lineages revealed that higher APESS scores were associated with higher infectivity (Figure 4). We used in silico and in vitro validation to reinforce our findings then used machine learning to make predictions on future developments (Figure 5). We created candidate sequences for evaluation and utilized machine learning in predictions (Figure 6).

We have added explanations to each section in Results that elucidate the objective of each section and how they connect with each other in the wider study.

In line 157, we have added ‘We examined the amino acid sequences of SARS-CoV-2 to make discoveries about biochemical properties.’ to clearly outline the objective of the subsection.

In line 207, we have improved the phrasing of the sentence.

In line 278, we stressed that ‘We developed APESS, an evaluation model to analyze viral sequences based on the nucleotide, amino acid, and protein structure properties.’ to properly define the purpose and background of APESS.

Please define abbreviations when they first appear.

We have added the full terms for the stated abbreviations in the relevant sections of the manuscript.

In line 107, we have added the proper abbreviation for Our World in Data (OWID).

In lines 143, 175, and 489 we have added the full term for Variants of Concern (VOCs).

In line 160, we have added the full term for Receptor Binding Motif (RBM).

**Reviewer #2 (Recommendations for the authors):**
(1) pg 9, line 51, full name of RBM should be declared.

We have added the full name of Receptor Binding Motif (RBM) to the appropriate section in the Abstract.

(2) How are the Variants of Concern (VOCs) defined?

Thank you for the comment and we apologize for the confusion. Variants of Concern as defined by the World Health Organization are specified in the Materials and Methods section. We have also added the full name for Variants of Concern (VOCs) when they are first mentioned in the Introduction and Results sections.

(3) pg 17, line 297. The purpose of using AI/ML to predict amino acid substitutions at specific locations is not clear. The VOCs and related mutation loci were already searched, so the AA substitution prediction step seems a little repetitive. Is it to create customized sequences? Also, if prediction (or probability) was made, some performance evaluation would be helpful.

Thank you for this feedback. The purpose of utilizing machine learning to make predictions about amino acid substitutions is to assess the possibility of amino acid substitutions occurring at specific locations. These potential amino acid substitutions were evaluated by APESS to have high scores, linking them to high infectivity. As the feedback suggests, amino acid substitutions in VOCs are researched but our prediction sought to ascertain the likelihood of amino acid substitutions that our evaluation model associated with infectivity. In the Results section in line 330, we assessed the probability of amino acid substitutions N460K and Q493R that the study found to be significant. The datasets that we utilized for these predictions are detailed in the Materials and Methods section in line 677.

The models we trained with machine learning predicted the probability of mutations based on samples in each group and their performance was evaluated by comparing the presence of mutations in the clades they diverged from. We have added the following sentences to line 330: “We used Accuracy, Precision, Recall, and F1 score to evaluate performance. All models showed high performance scores above 0.95 in Precision, Recall, and F1 score. For accuracy, XGBoost, scored above 0.89, exhibiting relatively high performance while LightGBM scored above 0.78.”

(4) pg 17, line 289. The objective of creating candidate lineages is not clear and would be helpful for the readers if its purpose is elaborated on. Since there are enough SARS-CoV-2 sequences, wouldn't it be more realistic and accurate to use those real sequences instead of creating them? Furthermore, the candidate lineages should be defined but they were missing in this section. This part made it a little difficult to follow the overall paper's logic.

The manuscript should have been clearer on what ‘candidate lineages’ signified, we apologize for the confusion. In line 314, we included the following sentences for clarity: ‘We introduced amino acid substitutions at specific locations in the SARS-CoV-2 backbone for the wildtype and VOCs. The amino acid substitutions were lysine (K), arginine (R), asparagine (N), serine (S), tyrosine (Y), and glycine (G). We then evaluated the infectivity of these candidate lineages with our evaluation model APESS.’

The purpose of creating candidate lineages in our study was to assess the effect of specific amino acid substitutions on the virus’ infectivity. The amino acid substitutions we evaluated were lysine (K), arginine (R), asparagine (N), serine (S), tyrosine (Y), and glycine (G). We determined that examining the introduction of specific amino acid substitutions to SARS-CoV-2 sequences would highlight the significance they had on infectivity. We have revised the paragraph in line 314 of the Results section to convey what we were doing.

(5) This study covers very detailed contents regarding lineages, mutations, and their effect on infectivity. It would be more readable if subsections could be added per group of investigation, especially in the results and discussion section.

In the Results section, we have emphasized the objective of each subsection and how they connect with one another for the overall goals of our study.

In line 157, we have added ‘We examined the amino acid sequences of SARS-CoV-2 to make discoveries about biochemical properties.’ to clearly outline the objective of the subsection.

In line 207, we have improved the phrasing of the sentence.

In line 278, we stressed that ‘We developed APESS, an evaluation model to analyze viral sequences based on the nucleotide, amino acid, and protein structure properties.’ to properly define the purpose and background of APESS.

We have made edits to the Discussion section to more clearly indicate subsections.

In line 389, we have added ‘In our investigation of various viruses’ to clearly indicate the background on other viruses.

In line 409, we added the sentence ‘We made discoveries on specific amino acid substitutions at positions.’ to indicate the subsection talking about N437R, N460K, and D467 mutations.

In line 471, we added the sentence ‘We created AIVE to feature our findings and analyses on an online platform.’ And modified the following sentence to better explain AIVE.

(6) pg 26, line 557. The criteria for the SCPSi scores were set to 0.9 and 0.1 by the proportion of the Omicron and Delta variants. How do other criteria affect the performance of the method?

Thank you for the question and check point. We used 0.9/0.1 for our initial criteria in our SCPS calculation. To determine how that affected performance, we have used 0.8/0.2 and 0.7/0.3 as the criteria.

After calculating APESS with different SCPS weights (0.9/0.1, 0.8/0/2, 0.7/0.3), we used a Gaussian Mixture Model (GMM) to compare how the groups were divided based on APESS. All three groups with different SCPS weights were determined to accurately reflect data patterns when they had four components.

When comparing parameter values, the group that used the original weights of 0.9 and 0.1 for SCPS showed the lowest values for variance and standard error across all four components. This indicates that each component was stable and clearly distinguishable from one another.

The group where the weights were adjusted to 0.7 and 0.3 for SCPS showed significantly higher variance and a large error for the G2 component. The distribution of each component was more widespread, signifying that the stability and reliability was lower.

The group where the weights were adjusted to 0.8 and 0.2 for SCPS was positioned between the two previous groups for finer data classification and reliability. However, the group notably lacked reliability when it came to the SE values for the G4 component.

Thus, the original model with 0.9 and 0.1 weight is the most reliable.

When the Gaussian Density for each group was plotted, the group with 0.9/0.1 SCPS weights showed the highest peak near 2 (G1), with a value of approximately 2. For the group with SCPS 0.8/0.2 weights, the highest peak appeared near 4.2 (G3), showing a high value around 14. For the group with SCPS 0.7/0.3 weights, the highest peak appeared near 3.7 (G3) showing a value around 5. The group with 0.9/0.1 SCPS weights exhibited a more uniform Gaussian distribution compared to the other two.

**Author response image 1. sa3fig1:** Superposition of Gaussian Densities for SCPS weight 0.9/0.1.

**Author response table 1. sa3table1:** Statistical values of the Superposition of Gaussian Densities for SCPS weight 0.9/0.1.

Component	mean	variance	se	weight
G1	2.0642	0.0034	0.058309519	0.4104
G2	1.8725	0.0778	0.278926514	0.0388
G3	0.0263	0.0056	0.074833148	0.2024
G4	1.6404	0	0	0.3484

**Author response image 2. sa3fig2:** Superposition of Gaussian Densities for SCPS weight 0.8/0.2.

**Author response table 2. sa3table2:** Statistical values of the Superposition of Gaussian Densities for SCPS weight 0.8/0.2.

Component	mean	variance	se	weight
G1	0	1.00E-06	0.001	0.191166653
G2	8.434051839	0.004907285	0.070052018	0.226271799
G3	4.210249767	1.00E-06	0.001	0.353961013
G4	6.758534282	2.463725537	1.569625923	0.228600536

**Author response image 3. sa3fig3:** Superposition of Gaussian Densities for SCPS weight 0.7/0.3.

**Author response table 3. sa3table3:** Statistical values of the Superposition of Gaussian Densities for SCPS weight 0.7/0.3.

Component	mean	variance	se	weight
G1	0	1.00E-06	0.001	0.19116666
G2	7.306207524	2.700755873	1.643397661	0.229009764
G3	3.683968546	1.00E-06	0.001	0.353987719
G4	9.478264495	0.011051658	0.105126865	0.225835857

(7) Overall, the approach is very detailed and realistic. Just curious if this approach would be also applicable to other viruses such as influenza.

We appreciate the insightful comments from the reviewer, and this is a direction we hope to take our research in the future. Our study focused on SARS-CoV-2 and the properties we discovered from the virus’ spike protein interacting with the host’s ACE2 receptor. In our investigation of other coronaviruses such as MERS-CoV, SARS-CoV-1 possesses a different structure and properties than these viruses as we have illustrated in Supplementary Figure 24. We had provided explanations about our investigation of other viruses in the Discussion section. In line 389, we have added ‘In our investigation of various viruses’ to better signpost this section.